# Phosphoproteomic profiling of T cell acute lymphoblastic leukemia reveals targetable kinases and combination treatment strategies

Valentina Cordo'[1], Mariska T. Meijer[1], Rico Hagelaar[1], Richard R. de Goeij-de Haas[2,3], Vera M. Poort[1], Alex A. Henneman[2,3], Sander R. Piersma[2,3], Thang V. Pham[2,3], Koichi Oshima[4], Adolfo A. Ferrando[4], Guido J. R. Zaman[5], Connie R. Jimenez[2,3,7] & Jules P. P. Meijerink[1,6,7 ✉]

Protein kinase inhibitors are amongst the most successful cancer treatments, but targetable kinases activated by genomic abnormalities are rare in T cell acute lymphoblastic leukemia. Nevertheless, kinases can be activated in the absence of genetic defects. Thus, phosphoproteomics can provide information on pathway activation and signaling networks that offer opportunities for targeted therapy. Here, we describe a mass spectrometry-based global phosphoproteomic profiling of 11 T cell acute lymphoblastic leukemia cell lines to identify targetable kinases. We report a comprehensive dataset consisting of 21,000 phosphosites on 4,896 phosphoproteins, including 217 kinases. We identify active Src-family kinases signaling as well as active cyclin-dependent kinases. We validate putative targets for therapy ex vivo and identify potential combination treatments, such as the inhibition of the INSR/IGF-1R axis to increase the sensitivity to dasatinib treatment. Ex vivo validation of selected drug combinations using patient-derived xenografts provides a *proof-of-concept* for phosphoproteomics-guided design of personalized treatments.

[1] Princess Máxima Center for Pediatric Oncology, Utrecht, The Netherlands. [2] OncoProteomics Laboratory, Cancer Center Amsterdam, Amsterdam University Medical Centers, VU University, Amsterdam, The Netherlands. [3] Department of Medical Oncology, Cancer Center Amsterdam, Amsterdam University Medical Centers, VU University, Amsterdam, The Netherlands. [4] Institute for Cancer Genetics, Columbia University Medical Center, New York, NY, USA. [5] Oncolines B.V., Oss, The Netherlands. [6] Present address: Acerta Pharma (member of the AstraZeneca group), Oss, The Netherlands. [7] These authors jointly supervised this work: Connie R. Jimenez, Jules P.P. Meijerink. ✉email: j.meijerink@prinsesmaximacentrum.nl

T cell acute lymphoblastic leukemia (T-ALL) is an aggressive malignancy arising from aberrant proliferation of immature T cell progenitors and accounts for about 15% of pediatric ALL cases[1]. The current risk-adapted, multiagent chemotherapeutic regimen has led to an overall survival rate exceeding 80%. Nevertheless, one out of five children with T-ALL will relapse within 4 years after the start of therapy. Further intensification of the current high-risk treatment protocols seems not feasible due to serious and even fatal detrimental side effects, such as toxicities and infections[2]. Relapsed T-ALL patients have a poor prognosis and are refractory towards further treatment. Therefore, the identification of novel therapeutic options for refractory/relapsed patients remains an urgent need. Thanks to extensive genome sequencing studies, the genetic drivers of T-ALL have been identified as developmental transcription factors that are ectopically expressed due to chromosomal rearrangements (reviewed in Belver et al.[3] and van der Zwet et al.[4]). Unlike other leukemias, genomic rearrangements involving kinase-coding genes are scarce in T-ALL. The most common aberration involving a kinase-coding gene is the *NUP214-ABL1* episomal amplification, which is found in less than 6% of T-ALL patients at diagnosis and is often detected only in minor leukemic sub-clones[5]. Recurrent activating mutations detected in kinase-coding genes or kinase regulators involve the PI3K-AKT axis (*AKT1*, *PIK3CD*, and *PIK3R1*), the JAK-STAT (*IL7R*, *JAK1*, and *JAK3* which is mutated in about 16% of T-ALL cases[6]), or the Ras signaling pathways (*PTPN11*, *NF1*, *N-RAS*, and *K-RAS*), while in some early T cell precursor (ETP)-ALL cases, Fms-like tyrosine kinase (*FLT3*) mutations and/or overexpression are found[7]. Additionally, other potentially druggable kinases reported for T-ALL include the JAK-family member tyrosine kinase 2 (TYK2) which can be activated either by rare *gain-of-function* mutations or IL-10 signaling[8,9], the cell cycle regulators Polo-like kinases (PLKs) and Aurora kinases (AURKs)[10,11], and the PIM1 kinase which can be upregulated by active IL-7 signaling, upon glucocorticoid-induced *IL7RA* expression, or in the presence of IL-7R pathway mutations[12,13].

Nevertheless, genomic-guided targeted therapies can show disappointing results due to the sub-clonal nature of these mutations (i.e., leukemia heterogeneity)[5,14]. The treatment pressure can drive the selection of minor resistant clones[15], induce the acquisition of novel mutations[16] or activate alternative feedback loops that drive therapy resistance. Leukemic cells rely on enhanced kinase signaling that promotes aberrant proliferation and survival. Protein kinases can be activated in the absence of gene fusions or mutations in their coding sequences. In fact, except for mutations in *JAK1/2* and *FLT3*, no other somatic mutation in tyrosine kinase-coding genes were found in 45 high-risk B-ALL cases, although the gene expression profiles indicated an active kinase signaling[17]. Therefore, proteome analyses can provide additional insights into active signaling pathways and kinases that could be exploited for targeted therapy. Mass spectrometry (MS)-based phosphoproteomics importantly contributed to the identification of signaling pathways and protein networks that can be targeted for cancer therapy[18–20]. Recently, Frejno and colleagues performed a large-scale proteome and phosphoproteome profiling of 125 cancer cell lines to create a proteomic activity landscape that can predict drug sensitivity in vitro[21]. Moreover, additional phosphoproteomic studies identified determinants of sensitivity to clinical kinase inhibitors in acute myeloid leukemia (AML) cell lines[22] and primary cells[23]. In the context of T-ALL, reverse phase protein array (RPPA)-based proteomic studies identified highly active signaling pathways in ETP-ALL such as the mTOR/STAT3 and LCK/calcineurin[24]. Additionally, Degryse and colleagues applied phosphoproteomics to investigate the

signaling pathways downstream of mutant *JAK3* in T-ALL[25]. Recently, Franciosa and colleagues used proteomic analyses to unravel the mechanisms of resistance to NOTCH1 inhibition in T-ALL[26]. Nevertheless, to our knowledge, no unbiased, MS-based phosphoproteomic study to predict drug sensitivity has hitherto been performed in T-ALL. Here, we present an exploratory MS-based, unbiased, global profiling of tyrosine, serine, and threonine phosphorylation in a panel of T-ALL cell lines and patient-derived xenografts (PDXs) to identify relevant kinase signaling and to predict novel dependencies. We validate highly active kinases as potential targets for therapy in vitro using both cell lines and PDX models. Furthermore, we demonstrate how the application of phosphoproteomics can guide the ex vivo identification of synergistic combination treatments, and the selection of the most appropriate therapeutical strategy for personalized medicine.

## Results

**Unbiased analysis of the global phosphoproteome in T-ALL cell lines.** To explore the phosphoproteome of human T-ALL, we performed global, unbiased mass spectrometry-based phosphoproteomic profiling of protein extracts from 11 T-ALL cell lines (Supplementary Table 1) as illustrated in Fig. 1a. Following phospho-tyrosine (pY) peptide immunoprecipitation, we identified about 3800 phosphosites while the titanium dioxide (TiO$_2$)-based enrichment yielded over 17,000 phosphosites. The identification of phospho-tyrosine peptides was notably higher for HSB-2 cells compared to the other cell lines (Supplementary Fig. 1a). This higher recovery correlates with an enhanced overall phospho-tyrosine signal in the unsupervised phosphopeptides cluster analysis, which was also confirmed by western blotting (Supplementary Fig. 1b, c).

To identify (hyper) active protein kinases that may be targeted by small-molecule inhibitors, the Integrative iNferred Kinase Activity (INKA) pipeline[27] was used to infer highly active kinases from phosphoproteomic data. This analysis pipeline provides a numerical single score as a proxy for kinases activity detected in a sample. The kinase ranking from the pY dataset revealed the broad activation of the Src-family kinases (SFKs) LCK, SRC, and FYN in all the cell lines analyzed, while other Src-family members such as ABL1, LYN, and FGR were detected only in specific lines, including PEER (ABL1), ALL-SIL (ABL1), MOLT-16 (LYN), LOUCY (LYN), and HPB-ALL (FGR) (Fig. 1b and Supplementary Figs. 1d and 2a–f). Three cell lines present a known genetic aberration that involves a kinase-coding gene, including the *TCRβ-LCK* translocation in the HSB-2 line and the *NUP214-ABL1* fusions in the cell lines ALL-SIL and PEER. Correspondingly, we identified LCK and ABL1 as the highest-ranking kinases in these three lines, respectively (Fig. 1b). Nevertheless, LCK shows high activation even in the absence of known genetic alterations in the remaining lines. Other active kinases identified include the cyclin-dependent kinases CDK1 and CDK2, the housekeeping kinases GSK3Bα/β, the insulin receptor (INSR) and the insulin-like growth factor receptor (IGF-1R) as illustrated in Fig. 1b.

The INKA ranking obtained from the TiO$_2$ dataset confirmed high activation of the cell cycle regulators CDK1 and CDK2 as a general hallmark for all lines. Additionally, INKA uncovered other potentially relevant kinases such as the dual-specificity kinase CLK1, the p21-activated kinases PAK1 and PAK2, and AKT1 (Fig. 1c and Supplementary Fig. 3). Interestingly, some cell lines showed a modest mTOR activity, while Jurkat cells had high MAPK/ERK activity, with MAPK1 and MAPK3 ranking 5 and 12, respectively in the TiO$_2$-INKA plot (Fig. 1c). To assess the reproducibility of our pipeline, two biological replicates for

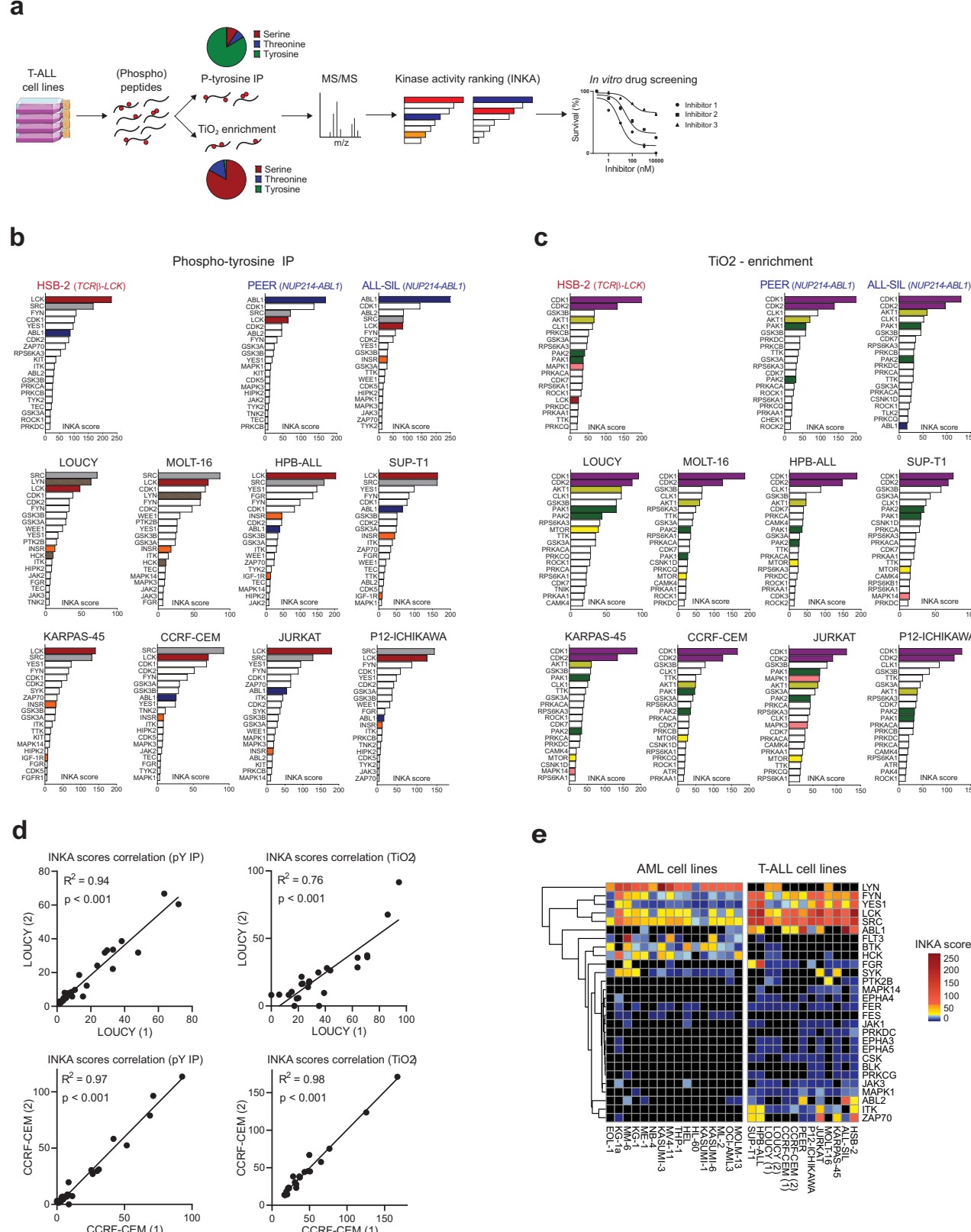

LOUCY and CCRF-CEM were included for both phospho-enrichment procedures. As illustrated in Fig. 1d, the biological duplicates showed high correlation ($R^2 \geq 0.76$) among INKA scores for the pY and the TiO$_2$ datasets. Similarly, a high correlation ($R^2 \geq 0.89$) was found between technical duplicates for the TiO$_2$ enrichment in eight other lines (Supplementary Fig. 4).

We then compared the T-ALL INKA scores with the INKA scores of a published AML phosphoproteomic dataset[22]. We identified differential (FDR < 0.1) kinase activities that characterize the myeloid and lymphoid lineages, further highlighting the relevance of the acquired phosphoproteome data. Both types of leukemia have high activation of SFKs. However, while T-ALL shows high

**Fig. 1 Phosphoproteomic profiling and INKA analysis identify active kinases in T-ALL. a** Experimental overview. Protein extracts from 11 T-ALL cell lines were enriched for phosphopeptides by anti-phosphotyrosine immunoprecipitation (IP) and titanium dioxide ($TiO_2$)-enrichment. Phosphorylated proteins were identified by liquid chromatography-tandem mass spectrometry (LC-MS/MS). Kinase activities were inferred and ranked using the INKA pipeline[27] and selected kinase inhibitors were tested in vitro. **b** Top20 INKA kinases inferred from the phospho-tyrosine (pY) dataset. Each bar plot illustrates the highest 20 active kinases in each cell line ranked on their INKA score. Red, LCK; blue, ABL1; gray, SRC; orange, INSR/IGF-1R; striped pattern, myeloid-lineage kinases (LYN and HCK). **c** Top20 INKA kinases inferred from the $TiO_2$ dataset. Each bar plot illustrates the highest 20 active kinases in each cell line ranked on their INKA score (each graph is representative of a technical duplicate). Purple, CDK1/2; dark green, PAK1/2; light green, AKT; yellow, mTOR; pink, MAPK; red, LCK; Blue: ABL1. **d** INKA scores correlation plots for biological duplicates in pY and $TiO_2$ datasets. Each plot shows the correlation of the INKA scores between biological duplicates for LOUCY and CCRF-CEM cell lines in both the pY and $TiO_2$ datasets (Pearson's correlation, two-sided Student's $t$-test, $p < 0.001$). **e** Heatmap illustrating significantly different (FDR < 0.1) kinases based on INKA scores in T-ALL (pY dataset) and AML (pY dataset[22]); cell lines. Source data are provided as a Source Data file.

activity of LCK, SRC, ABL1, YES1, and FYN, AML cells show activation of LYN and HCK only (Fig. 1e). Interestingly, the ETP-ALL-like cell line LOUCY and the *LMO2*-rearranged MOLT-16 cells show activation of LYN and HCK as well (Fig. 1b, e), indicating that subsets of T-ALL cells can present myeloid-like signaling features, along with the expression of known immature markers such as CD34. Furthermore, we could identify other subtype-specific kinases such as the Bruton Tyrosine Kinase BTK and FLT3 for AML, while ITK and ZAP70 were identified for T-ALL (Fig. 1e). These activities reflect active signaling pathways and can arise independently from known activating mutations in kinase-coding genes in these lines. Therefore, our data give a comprehensive overview of kinase activation in T-ALL cell line models and points to several potentially targetable activities that could represent novel leukemia vulnerabilities.

**The CDK1/2 inhibitor milciclib effectively induces G1-cell cycle arrest in T-ALL cell lines in vitro.** Based on the kinases identified in our phosphoproteome profiling study, we tested the sensitivity of the cell lines to multiple kinase inhibitors in vitro (Supplementary Table 2) to uncover signaling dependencies that can be exploited for targeted therapy. Despite the high ranking of CLK1 (Top7 $TiO_2$-INKA ranking in every cell line, Fig. 1c), the CLK1 inhibitor TG-003 had only limited cytotoxic efficacy in these lines (Supplementary Fig. 5a). In addition, the PAK1/2 inhibitor FRAX597 showed some effects in ALL-SIL and HSB-2 cells, with $IC_{50}$ values around 400 nM, but was less effective in the other cell lines ($IC_{50}$ values above 1 μM; Supplementary Fig. 5b). Therefore, CLK1 or PAK1/2 inhibition alone does not suffice to effectively impair T-ALL cell survival. Similar results were obtained with the AKT inhibitor ipatasertib (Supplementary Fig. 5c). Despite the low ranking of mTOR activity, the mTOR inhibitor sirolimus significantly reduced cell viability at nanomolar concentrations in five cell lines (Supplementary Fig. 5d). Of note, no effect was seen upon mTOR inhibition in healthy human thymocytes (Supplementary Fig. 5e). Jurkat cells showed high ERK activity (MAKP1 and MAPK3), but cells were insensitive to the MEK inhibitor selumetinib, indicating that Jurkat cells do not essentially depend on ERK signaling for survival (Supplementary Fig. 5f). In the $TiO_2$-INKA dataset, CDK1 and CDK2 were the top2 ranking kinases in all cell lines analyzed (Fig. 1c). Interestingly, the CDK1/2 inhibitor milciclib induced an effective reduction of cell survival in all lines tested, with $IC_{50}$ values between 50 nM and 1 μM (Fig. 2a). To investigate the mechanism of action, we performed cell cycle analysis that highlighted an induction of G1-arrest upon milciclib treatment (Fig. 2b). Annexin V/PI staining revealed that induction of apoptosis only occurs at higher drug concentrations (1 μM) with a drastic effect in HSB-2 cells and a less pronounced effect in the remaining lines, indicating that milciclib mainly acts as a cytostatic drug (Fig. 2c). To investigate potential cytotoxic mechanisms in HSB-2 cells, we looked for possible off-target effects of milciclib. Thus, we

browsed the publicly available chemical proteomic database ProteomicsDB[28,29] (https://www.proteomicsdb.org/) and found that milciclib can also inhibit LCK ($EC_{50}$ 1.4 μM). We validated the reduced phosphorylation of SFKs, including LCK, upon milciclib treatment by western blot (Fig. 2d). Since HSB-2 cells present a driver *TCRβ-LCK* translocation that induces ectopic LCK expression, cells highly depend on LCK signaling for their survival. Therefore, milciclib efficacy in T-ALL could be higher in LCK-dependent cells. Eventually, we confirmed milciclib treatment efficacy in four T-ALL patient-derived xenografts treated ex vivo (Supplementary Fig. 5g, h).

**T-ALL cell lines show limited sensitivity to SRC-family kinases inhibition in vitro.** To further investigate the potential of LCK as therapeutical target, we analyzed the pY dataset. A predominant role of Src-family members emerged among the detected tyrosine kinases, in particular LCK and SRC (Fig. 1b). Despite the ranking of SRC and LCK as Top2 pY-kinases in most lines (Fig. 1b and Supplementary Fig. 2a, b), only HSB-2 and ALL-SIL cells were highly sensitive to ATP-competitive SRC/ABL inhibitors (dasatinib, ponatinib, bosutinib, nilotinib, and imatinib) with $IC_{50}$ values below 100 nM. These lines are characterized by a *TCRβ-LCK* translocation or a *NUP214-ABL1* fusion, respectively (Fig. 3a). The PEER cell line, also described as a *NUP214-ABL1* fusion-positive line, had a low sensitivity to SKFs inhibition ($IC_{50}$ for SRC/ABL inhibitors above 1 μM except for ponatinib, 834 nM, Fig. 3a). The remaining cell lines responded to increasing doses of SRC/ABL inhibitors, but at concentrations beyond the clinically relevant concentrations that are achieved in patient's plasma (Supplementary Fig. 6a–e). Similar results were obtained with the LCK inhibitor A-420983 (Fig. 3a). Given the broad activation of SFKs detected, we then investigated the effects of dasatinib treatment. Western blotting confirmed high expression of both SRC and LCK in the cell line panel (Fig. 3b), and the lines with the highest LCK and SRC expression were used to investigate the effect of dasatinib in vitro. Dasatinib treatment (100 nM) for three days induced an effective decrease of phospho-LCK and phospho-SRC in all these lines as well as an apparent down-regulation of total LCK expression (Fig. 3c), without affecting the cell viability (Fig. 3a and Supplementary Fig. 6f) while HSB-2 cells show induction of apoptosis already after 16 h of dasatinib treatment (Supplementary Fig. 6g). Thus, our data suggest that LCK and SRC are highly active in T-ALL, but the pharmacological inhibition of these activities is only effective in HSB-2 and ALL-SIL that harbor rearrangements in *LCK* or *NUP214-ABL1*, respectively. Remarkably, dasatinib-resistant T-ALL cell lines also downregulate LCK upon dasatinib treatment, indicating their limited dependency on LCK activity for their survival. Therefore, the inhibition of LCK alone seems not a universal effective treatment for every T-ALL, possibly due to the activation of alternative signaling routes.

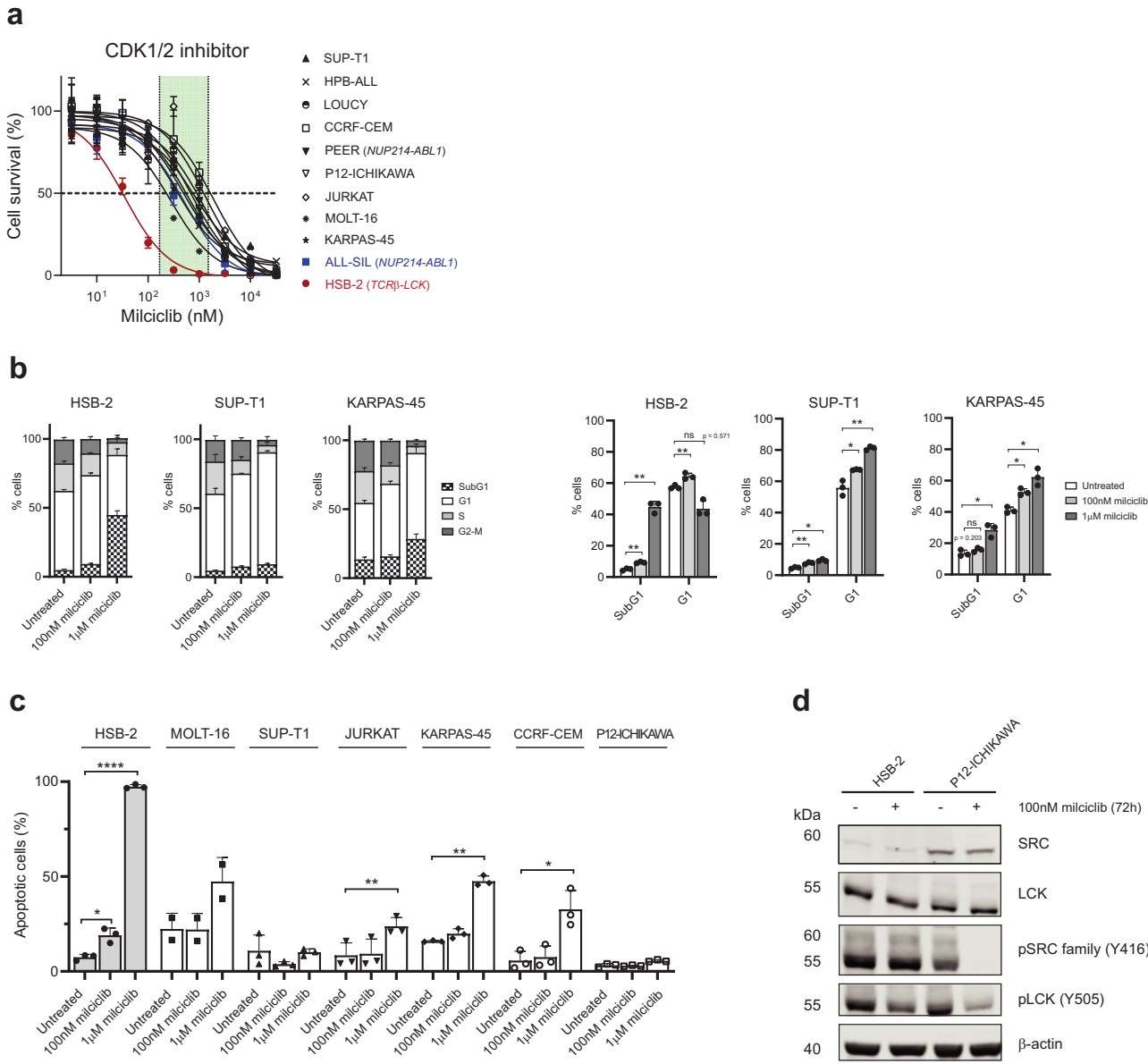

**Fig. 2 The CDK1/2 inhibitor milciclib induces G1 arrest in T-ALL cell lines. a** Dose-response curves of milciclib treatment in 11 T-ALL cell lines. Cells were treated with increasing concentrations of milciclib (3.2 nM–32 µM range) in triplicate and cell viability was assessed after 72 h using the colorimetric MTT assay. Cell survival was calculated in comparison to the untreated control. Each point represents the mean and standard deviation of the triplicate. The green box shows the corresponding clinical concentration range of milciclib used in patients enrolled in clinical trials[51]. **b** Cell cycle analysis upon milciclib treatment. Cells were treated with either 100 nM or 1 µM milciclib for 72 h and cell cycle analysis was performed via Hoechst-DNA staining and FACS analysis. The graphs show the average and standard deviation of three independent experiments. Significance was determined using a paired, two-tailed Student's t-test and annotated as "ns" (not significant, $p \geq 0.05$), * ($p < 0.05$), ** ($p < 0.01$). **c** Detection of apoptotic cells upon milciclib treatment. Cells were treated with either 100 nM or 1 µM milciclib for 72 h and Annexin V/ Propidium Iodide (PI) staining was used to detect apoptosis. Apoptotic cells were identified as the sum of the Annexin V+ cells and Annexin V+/PI+ cells. The percentage of apoptotic cells is calculated compared to untreated control cells. The graphs show the average and standard deviation of three independent experiments. Significance was determined using a paired, two-tailed Student's t-test and annotated as *($p < 0.05$), **($p < 0.01$), ****($p < 0.0001$). If not annotated, the results were not significant ($p \geq 0.05$). **d** Western blot analysis upon milciclib treatment. HSB-2 cells and P12-ICHIKAWA cells were treated with 100 nM milciclib for 72 h and 20 µg of protein was used for each condition. The image is representative of three independent experiments. Source data are provided as a Source Data file.

**Inhibition of the INSR/IGF-1R axis sensitizes T-ALL cells to dasatinib treatment in vitro**. To further investigate the role and the possible targeting of Src-family kinases in T-ALL, we looked for possible upstream kinases or receptors that can explain the activation of LCK and the other SFKs. Interestingly, INSR and IGF-1R were amongst the top 20-activated kinases in the pY-INKA profiles for nine out 11 lines (Figs. 1b and 4a). Therefore, we tested the sensitivity of these lines to the INSR/IGF-1R

inhibitor BMS-754807. ALL-SIL, HPB-ALL, and MOLT-16 demonstrated sensitivity to single BMS-754807 treatment with IC50 values below 300 nM while most of the other cell lines had IC50 values around 1 µM (Fig. 4b). SUP-T1 and Jurkat cells were resistant to BMS-754807 treatment (IC50 approximately 10 µM) despite the predicted INSR/IGF-1R activity (Fig. 4b) possibly due to alternative survival signaling pathways. Since SRC can act as signal transducer downstream of several membrane-receptors, we

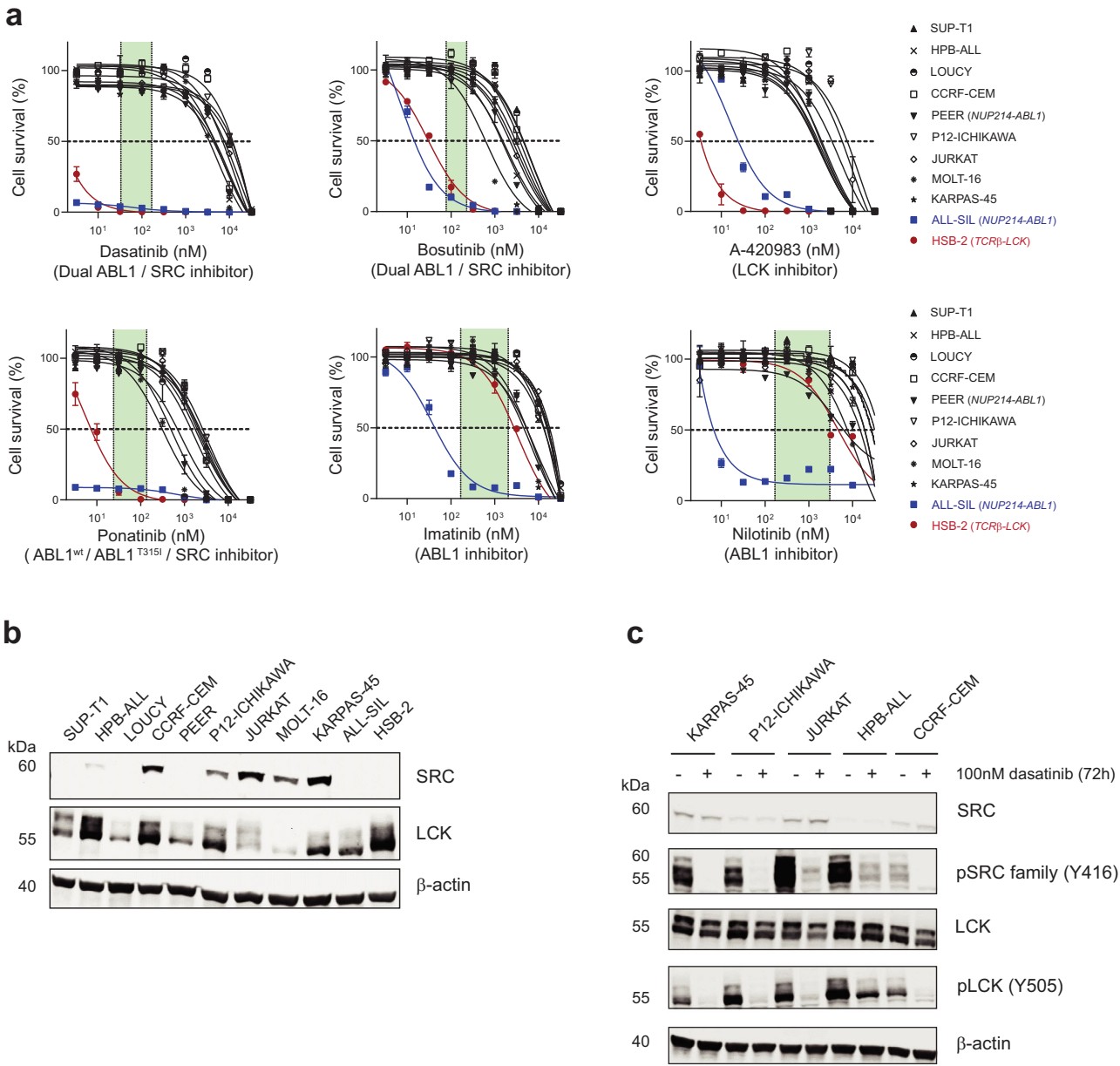

**Fig. 3 T-ALL cell lines show limited sensitivity to SFKs inhibition in vitro. a** Dose-response curves of SFKs inhibitors treatment in 11 T-ALL cell lines. Cells were treated with increasing concentrations of different SFKs inhibitors: dasatinib, bosutinib, A-420983, ponatinib, imatinib and nilotinib (3.2 nM–32 μM range) in duplo and cell viability was assessed after 72 h using the ATPLite assay (PerkinElmer). Cell survival was calculated in comparison to the untreated control. Each point represents the mean and standard deviation of the duplicate. The green box shows the range of clinical concentrations either derived from pharmacodynamics studies or based on drug dosages used in current clinical trials[52–55]. **b** Western blot analysis showing LCK and SRC expression in untreated T-ALL cell lines. 30 μg of protein input was used for each sample. The image is representative of two independent experiments. **c** Western blot analysis upon dasatinib treatment. Cell lines expressing high levels of LCK and/or SRC were treated with 100 nM dasatinib for 72 h and 30 μg of protein was used per sample. The image is representative of two independent experiments. Source data are provided as a Source Data file.

questioned whether lines with active SRC, LCK, and INSR/IGF-1R signaling could benefit from combined SFKs and INSR/IGF-1R signaling inhibition. Therefore, we evaluated the effect of the addition of a low BMS-754807 dose ($IC_{20}$) to the dasatinib treatment in vitro. The lowest dose of dasatinib tested (3.2 nM) in combination with low concentrations of BMS-754807 (30–300 nM) showed a synergistic and superior effect compared to the single treatments (Fig. 4c, d). For SUP-T1 cells, the addition of 30 nM BMS-754807 to the dasatinib treatment strongly enhanced the cytotoxic effects (CI < 0.1; Fig. 4d). As validation of the combined treatment strategy, two other INSR/IGF-1R inhibitors were tested in SUP-T1 cells, linsitinib (OSI-906) and GSK-

4529A, respectively. Like the BMS-754807 single treatment, SUP-T1 cells showed low sensitivity to both INSR/IGF-1R inhibitors as monotherapy (Supplementary Fig. 7a, b). However, the combination of a low dose ($IC_{20}$) of linsitinib or GSK-4529A to the dasatinib treatment in vitro confirmed the synergism of simultaneous SRC/LCK and INSR/IGF-1R inhibition (Fig. 4e). To investigate the mechanisms for the synergistic efficacy of this drug combination, we performed western blotting using SUP-T1 cells after treatment with dasatinib, BMS-754807, or their combination. Single 3.2 nM dasatinib treatment reduced the phosphorylation of SFKs and ERK1/2 while single 30 nM BMS-754807 treatment reduced the phosphorylation of IGF-1Rβ as well as

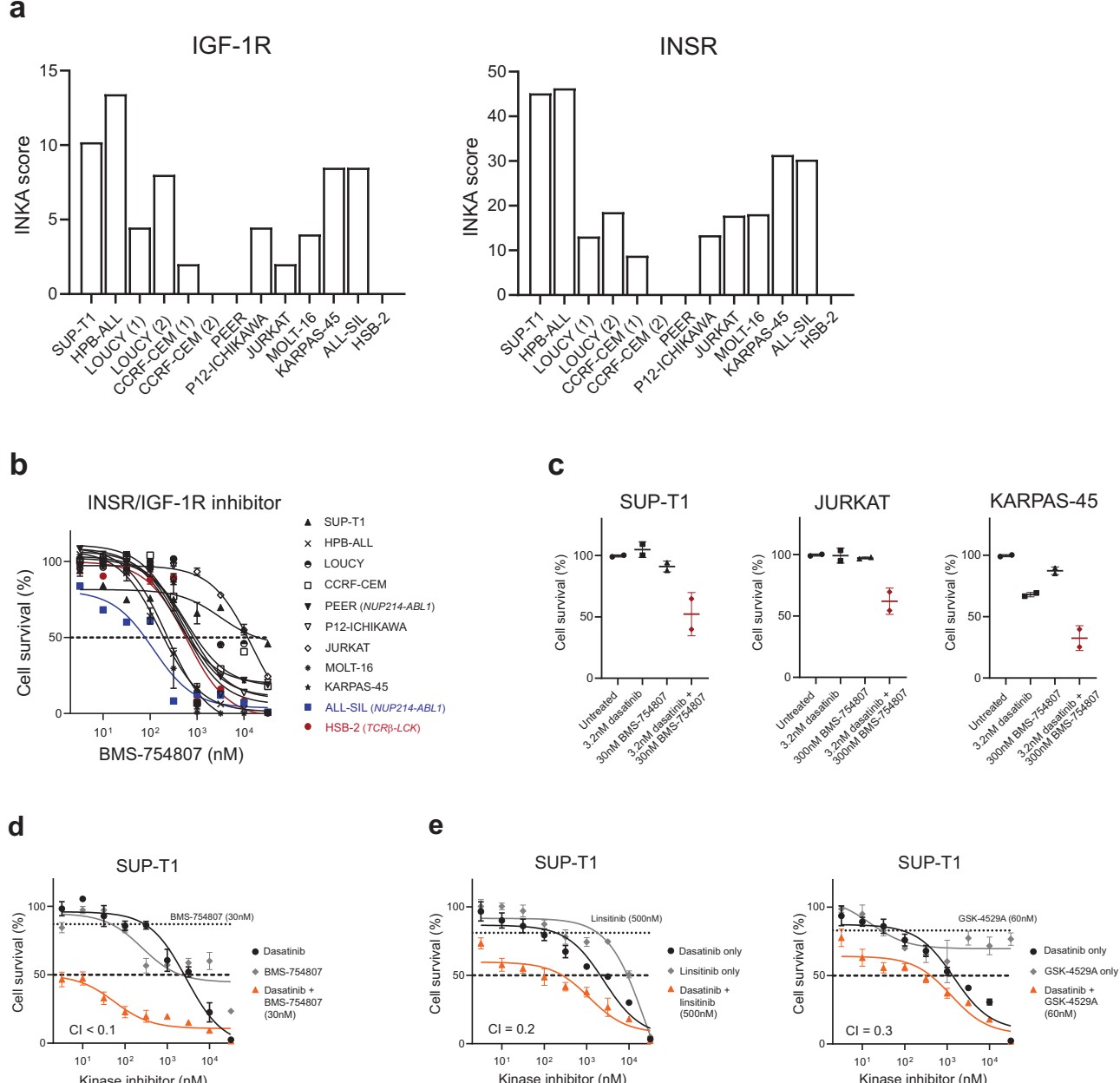

**Fig. 4 IGF-1R inhibition can sensitize cells to dasatinib treatment. a** Bar plots of the INKA scores inferred from the pY dataset for IGF-1R and INSR in each cell line. **b** Dose-response curves of BMS-754807 treatment in 11 T-ALL cell lines. Cells were treated with increasing concentrations of the INSR/IGF-1R inhibitor BMS-754807 (3.2 nM–32 μM range) in duplicates. Cell survival after 72 h was calculated in comparison to the untreated control. Each point represents the mean and the standard deviation of the duplicates. **c** Sensitivity to dasatinib, BMS-754807 and the combination of 3.2 nM dasatinib with the IC$_{20}$ concentration of BMS-754807 (30–300 nM) for SUP-T1, Jurkat and KARPAS-45 cells. Data is shown as mean and standard deviation of two independent experiments performed in triplicate. **d** Dose-response curves of dasatinib, BMS-754807 and combination of dasatinib and a fixed concentration of BMS-754807 treatment in SUP-T1 cells. Cells were treated for 72 h with increasing concentrations of either dasatinib or BMS-754807 alone or with a combination of dasatinib (3.2 nM–32 μM range) with a fixed concentration of BMS-754807 (30 nM, corresponding to the IC$_{20}$ of the single treatment for SUP-T1 cells indicated by the dotted line), in triplicate. Cell survival was calculated in comparison to the untreated control. The graph is representative of three independent experiments. CI combination index. **e** Dose-response curves of dasatinib, linsitinib, GSK-4524A and combination of dasatinib and a fixed concentration of either linsitinib or GSK-4529A treatment in SUP-T1 cells. Cells were treated for 72 h with increasing concentrations of dasatinib or linsitinib/GSK-4529A alone or with a combination of dasatinib (3.2 nM–32 μM range) with a fixed concentration of linsitinib (500 nM, corresponding to the IC$_{20}$ of the single treatment for SUP-T1 cells indicated by the dotted line) or GSK-4529A (60 nM, corresponding to the IC$_{20}$ of the single treatment for SUP-T1 cells indicated by the dotted line), in triplicate. Cell survival was calculated in comparison to the untreated control. The graph is representative of three independent experiments. CI combination index. Source data are provided as a Source Data file.

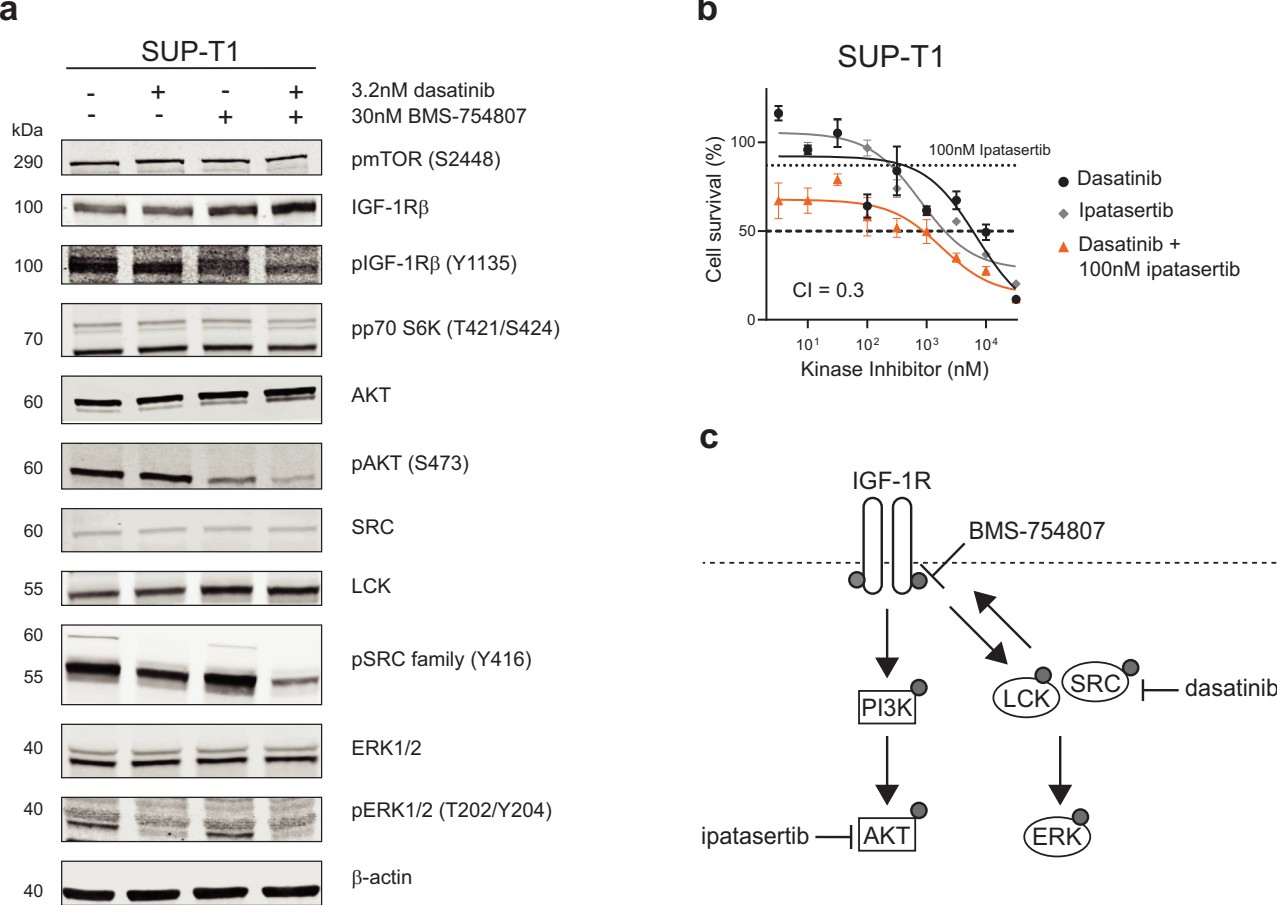

**Fig. 5 Concomitant inhibition of SFKs and IGF-1R inhibits AKT. a** Western blot of SUP-T1 cells treated with dasatinib, BMS-754807 or combination treatment. Cells were incubated for 72 h with 3.2 nM dasatinib, 30 nM BMS-754807 or combined 3.2 nM dasatinib and 30 nM BMS-754807. 50 μg of protein input was used for each sample. The image is representative of two independent experiments. **b** Dose-response curves of dasatinib, ipatasertib, and combination of dasatinib and a fixed concentration of ipatasertib treatment in SUP-T1 cells. Cells were treated for 72 h with increasing concentrations of either dasatinib or ipatasertib alone or with a combination of dasatinib (3.2 nM–32 μM range) with a fixed concentration of ipatasertib (100 nM, corresponding to the $IC_{20}$ of the single treatment for SUP-T1 cells indicated by the dotted line), in triplicate. The data is presented as mean and standard deviation of the mean. The graph is representative of three independent experiments. CI combination index. Source data are provided as a Source Data file. **c** Graphical summary of the potential targeting of the INSR/IGF-1R axis and the SFKs signaling.

AKT (S473) (Fig. 5a). No effect was detected on phospho-mTOR or phospho-p70 S6 kinase upon BMS-754807 treatment (Fig. 5a), indicating that the IGF-1R signaling converges mainly on AKT. Interestingly, upon IGF-1R inhibition, a slight increase in phosphorylation of SFKs was noticed (phospho-SRC family Y416), further pointing to a potential cross-talk between IGF-1R and SFKs. In fact, the combination of 3.2 nM dasatinib and 30 nM BMS-754807 showed a further decrease in the activation of IGF-1R (phospho-IGF-1Rβ Y1135), phospho-AKT (S473), and phospho-SFKs (Fig. 5a). To further validate a role for AKT in IGF-1R and SFKs signaling, we tested the cytotoxic effects of the ATP-competitive AKT inhibitor ipatasertib and its combination with dasatinib treatment. Addition of 100 nM ipatasertib to the dasatinib treatment enhanced the cytotoxic effects (CI = 0.3; Fig. 5b). Thus, our data show that co-targeting of activated IGF-1R/AKT and SFKs (summarized in Fig. 5c) can extend the potential usage of dasatinib in T-ALL.

**INKA-guided ex vivo drug screenings identify synergistic combinations in T-ALL patient-derived xenografts**. To validate our approach, we performed phosphotyrosine proteome profiling followed by INKA prediction of active kinases using human T-ALL blasts that were obtained from four different murine

patient-derived xenografts (Fig. 6a). INKA prediction of active tyrosine kinases highlighted SFKs activation (LCK, SRC, FYN, and YES1) in all the PDXs analyzed, as well as activation of the INSR/IGF-1R axis in two PDXs (PDX-01 and PDX-02) and to a lesser extent in PDX-04 that presented only INSR activity with a low ranking (Fig. 6b). All PDXs showed sensitivity to dasatinib treatment ex vivo with $IC_{50}$ values lower than 100 nM while PDX-01 showed also high sensitivity to the BMS-754807 single treatment with an $IC_{50}$ of 234 nM (Fig. 6c). To further validate the SFKs and INSR/IGF-1R combined inhibition as a putative treatment strategy, blasts obtained from the four different T-ALL PDXs were treated with either dasatinib, BMS-754807 or the combination of both drugs ex vivo using a 10-by-10 drug combination matrix as illustrated in Fig. 6a. Zero-Interaction Potency (ZIP) analysis[30] of the drug matrix identified synergy between dasatinib and BMS-754807 treatment in one out of four PDXs (PDX-02) already at nanomolar concentrations, as illustrated in Fig. 6d. PDX-01 showed already high sensitivity to both single treatments (Fig. 6c) with the drug combination treatment yielding only an additive effect (ZIP synergy scores lower than 10). However, PDX-02 which had lower sensitivity to BMS-754807 single treatment ($IC_{50}$ 675 nM), showed high synergy upon combined dasatinib and BMS-754807 treatment, indicating that

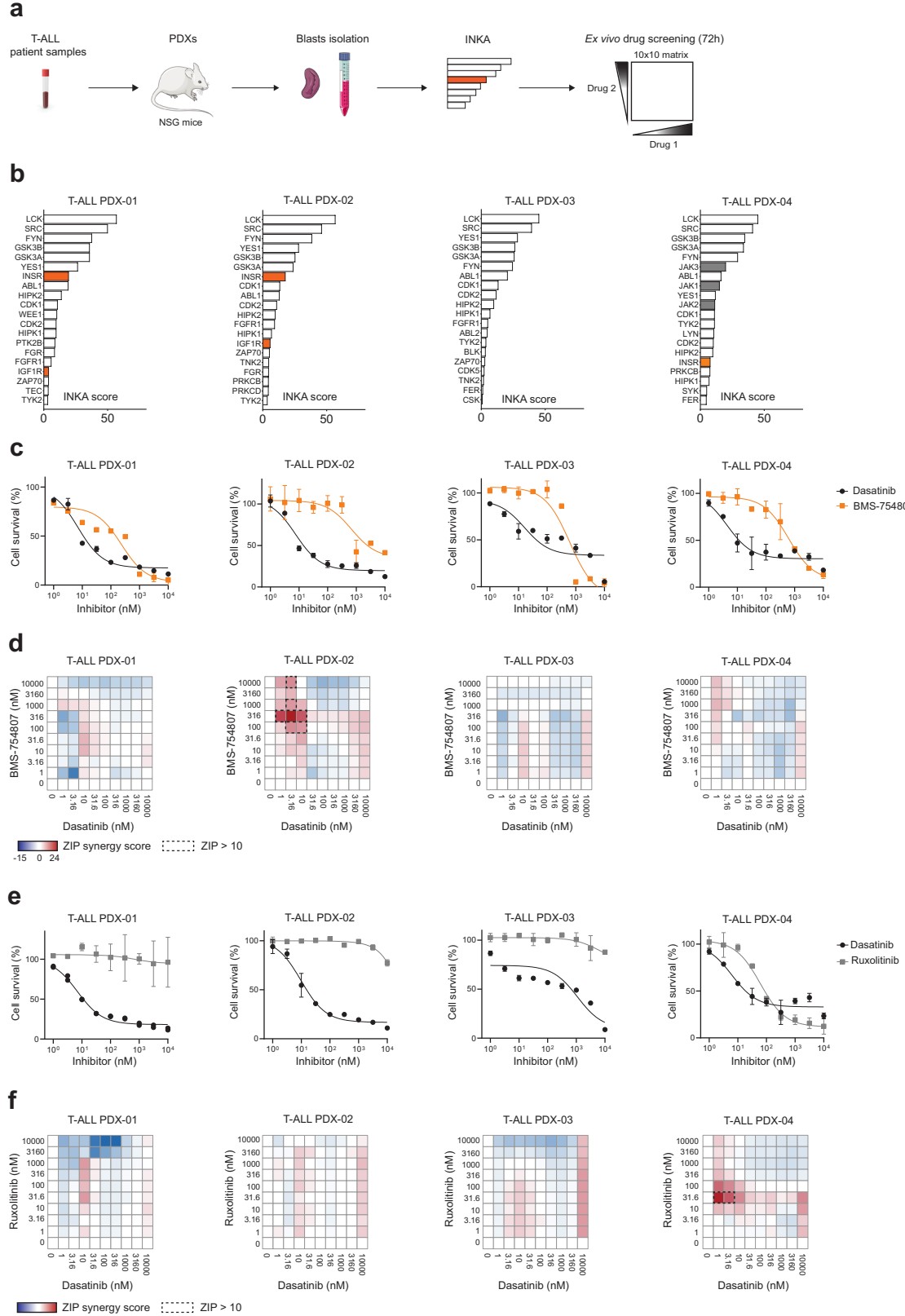

an active INSR/IGF-1R axis is a targetable vulnerability. Importantly, none of the PDXs carried any somatic mutation in the *INSR* or *IGF1R* gene (Supplementary Data 1), highlighting the power of phosphoproteomics in uncovering non-genomic targets for therapy. Consistent with a lack of INSR/IGF-1R activity (Fig. 6b), PDX-03 did not benefit from the combination

treatment (Fig. 6d). For PDX-04, INKA analysis predicted only INSR activity with a low ranking (16/20; Fig. 6b) indicating that INSR is not one of the dominant activities and thus explaining the lack of synergy upon combination treatment. Nevertheless, PDX-04 presented high Janus kinases (JAKs) activity which was absent in the other PDXs (Fig. 6b). Interestingly, the high JAK3 activity

**Fig. 6 INKA prediction of active kinases identifies synergistic combinations in patient-derived xenografts. a** PDXs were obtained from T-ALL primary cells expanded in NSG mice. Purified human blasts were used for phosphoproteomics, INKA analyses, and ex vivo drug screening using a 10-by-10 drug concentrations matrix. **b** Top20 INKA kinases from the phosphotyrosine (pY) dataset. Each bar plot illustrates the highest 20 active kinases in four PDXs ranked on their INKA score. Orange, INSR/IGF-1R; gray, JAKs; **c** Dose-response curves of dasatinib and BMS-754807 in T-ALL PDX cells (T-ALL PDX-01 to 04). Cells were treated for 72 h with increasing concentrations of either dasatinib or BMS-754807 (1 nM–10 μM range) and viability was calculated in relation to untreated control cells (DMSO only). Each point represents the mean and standard deviation of the duplicate. **d** Zero-Interaction Potency (ZIP) synergy scores for the combination of dasatinib and BMS-754807 in a 1 nM–10 μM concentration range. Cells were treated with either one of the single drugs or a drug combination for 72 h in duplicate. Cell survival was calculated in comparison to untreated cells (DMSO only). ZIP values lower than 0 indicate an antagonistic effect of the drug combination (blue), values between 0 and 10 indicate an additive effect (white to light red) while values above 10 (corresponding to a deviation from the reference model above 10%) indicate synergy (dark red and outside black dashed line). Each drug screening was performed in duplicate. **e** Dose-response curves of dasatinib and ruxolitinib in T-ALL PDX cells. Cells were treated for 72 h with increasing concentrations of either dasatinib or ruxolitinib (1 nM–10 μM range) and viability was calculated in relation to untreated control cells (DMSO only). Each point represents the mean and standard deviation of the duplicate. **f** ZIP synergy scores for the combination of dasatinib and ruxolitinib in a 1 nM–10 μM concentration range. Cells were treated with either one of the single drugs or a drug combination for 72 h in duplicate. Relative viability vas calculated in comparison to untreated cells (DMSO only). Each drug screening was performed in duplicate. Source data are provided as a Source Data file.

correlated with the presence of an activating $JAK3^{M511I}$ mutation (Supplementary Data 1). Therefore, we tested whether the combination of the JAK inhibitor ruxolitinib with dasatinib could be an effective treatment option for this specific T-ALL case. As shown in Fig. 6e, PDX-04 was the only sample sensitive to ruxolitinib ex vivo ($IC_{50} < 100$ nM) while the other PDXs remained completely insensitive to the treatment ($IC_{50} > 10$ μM). Moreover, ZIP analysis of the drug combination matrix identified synergy between ruxolitinib and dasatinib only when high JAK activity was predicted (PDX-04, Fig. 6f). Thus, the prediction of highly active kinases from phosphoproteomic data can guide the ex vivo evaluation of effective drug combination treatments, which can differ from patient to patient.

## Discussion
The identification of new targeted drugs is urgently needed to prevent relapses, overcome therapy resistance, and avoid excessive toxicities for T-ALL patients. Genomic-guided therapy has thus far not led to the wide implementation of targeted agents in T-ALL[31,32]. However, protein analyses can provide useful insights for the identification of cellular dependencies that translates into targetable leukemia vulnerabilities[4]. In this study, we performed an exploratory, global, unbiased phosphoproteome profiling to identify targetable kinases in T-ALL and to establish a proteome platform that can complement genomic analyses for the investigation of leukemia dependencies. In our initial analysis of 11 T-ALL cell lines, we identified highly active tyrosine kinases (LCK, SRC, FYN, YES1, LYN, INSR, and IGF-1R) as well as serine/threonine kinases (CDK1/2, AKT, and PAK1/2). Moreover, the comparison of the phosphoproteomes of T-ALL and AML revealed active kinases that reflect differences in their hematopoietic lineages of origin, independently of the presence of known signaling mutations, highlighting the additional value of MS-based phosphoproteome profiling. Next, we screened several clinical kinase inhibitors in vitro and found that the CDK1/CDK2 inhibitor milciclib has a cytostatic effect in T-ALL cells. Milciclib is under clinical investigation for the treatment of thymoma and hepatocellular carcinoma[33]. Currently, clinical studies are investigating other CDKs inhibitors (ribociclib and palbociclib) for the treatment of relapsed T-ALL[12] and milciclib may be an additional treatment option.

Cells with *ABL1* or *LCK* driving oncogenic aberrations showed high sensitivity to SRC/ABL inhibitors, including dasatinib, while the remaining cell lines had a limited response, despite the prediction of highly active Src-family members. Therefore, elevated LCK activity seems not to translate into cellular dependency in all T-ALL specimens. In 2017, Frismantas and colleagues showed that up to 30% of T-ALLs were sensitive to dasatinib ex vivo in

the absence of *ABL1* abnormalities. This dasatinib-responsiveness correlated with higher levels of phosphorylated SRC in sensitive cells[34]. In line with this previous study, our PDX models showed high sensitivity to dasatinib ex vivo in the absence of *LCK* or *SRC* mutation*s* (Supplementary Data 1), underscoring the potential use of this drug for T-ALL treatment. Recently, a pharmacogenomic study on pediatric T-ALL identified LCK, but not SRC, as driver of dasatinib sensitivity in up to 40% of pediatric T-ALL cases. The observed LCK activity correlated with pre-TCR signaling and relatively more mature developmental stages (TAL/LMO)[35], while immature ETP-ALL cells were less sensitive to dasatinib. In our study, the immature T-ALL cell line LOUCY indeed showed lower LCK activity but increased activation of myeloid kinases such as LYN and HCK. Given the lower response to dasatinib for most of the cell lines in our panel, possibly due to the presence of alternative escape signaling routes, we found that co-inhibition of the INSR/IGF-1R axis and SFKs was strongly synergistic. We provided evidence that the INSR/IGF-1R axis is active in most T-ALL cell lines and that the pharmacological inhibition of IGF-1R sensitizes T-ALL cells to dasatinib treatment, indicating important cross-talks between INSR/IGF-1R and SFKs. These results are in line with previous studies that identified INSR/IGF-1R activation as a bypass mechanism in solid tumors with intrinsic resistance to tyrosine kinase inhibitors[36–38]. Moreover, preclinical investigations showed that a subset of T-ALL cells is sensitive to INSR/IGF-1R inhibition without presenting any activating mutations in these receptor kinase-coding genes[39,40]. We validated the targeting of active INSR/IGF-1R signaling either as single treatment or in combination with dasatinib in two PDX models. In line with previous observations, both PDXs did not carry any somatic mutation in *INSR* or *IGF1R*. The lack of mutations that could explain the active INSR/IGF-1R signaling underscores the role of phosphoproteomics in highlighting relevant signaling nodes which would have not been uncovered via genomic analyses. Interestingly, Gocho and colleagues showed modulation of INSR activity upon dasatinib treatment in dasatinib-sensitive T-ALL patient-derived murine xenografts[35], further strengthening the observation that the INSR/IGF-1R and SFKs signaling can be interconnected and can mutually affect each other, as summarized in Fig. 5c, and illustrated in kinase-substrate relation networks in Supplementary Fig. 8. Two studies highlighted the role of dendritic cells and tumor-associated myeloid cells in supporting T-ALL growth in stromal niches via IGF-1R activation[41,42], emphasizing the relevance of this signaling pathway in the pathobiology of T-ALL. The tumor niche can provide a protective microenvironment, and therefore the simultaneous blocking of IGF-1R and SFKs signaling should be further investigated for T-ALL patients.

Furthermore, since several studies highlighted a role for LCK in supporting resistance to glucocorticoids in T-ALL[43,44], targeting LCK activation could provide additional benefits to other combination therapies. Multiple clinical trials are investigating the JAK inhibitor ruxolitinib for the treatment of T-ALL in the presence of *JAK* mutations[12]. Here, we show that ruxolitinib treatment is effective ex vivo in T-ALL cells with elevated JAK kinase activity. Interestingly, the elevated JAK3 activity correlated with the presence of an activating *JAK3* mutation, highlighting that driving oncogenic aberrations can also be detected at the signaling level. Moreover, ruxolitinib can synergize with dasatinib treatment in JAK-activated and SFKs-activated cells, indicating another putative combinatorial strategy for selected T-ALL cases. Therefore, our phosphoproteomic profiling provides a platform for the investigation of combinatorial treatments and for the identification of non-genomic leukemia dependencies. Such dependencies can be further exploited as leukemia vulnerabilities for personalized treatment.

INKA-based selection of (combination) treatments has been already validated in the context of solid tumors[27] and acute myeloid leukemia[22,45], underscoring the functional value of the pipeline. Future T-ALL studies should include an in vitro screening platform that can allow blasts proliferation ex vivo to study drugs affecting the cell cycle, as well as an extended PDXs cohort comprising all the different T-ALL subtypes. Further in vivo investigations of selected drug combinations should address not only the efficacy and tolerability of these treatments (i.e., toxicities) but also the role of the microenvironmental niches that can support blasts growth and survival. Such investigations could allow the direct translation of the preclinical findings to the clinical settings. In conclusion, we provide evidence that phosphoproteomics can guide the selection of targets for ex vivo drug screening to evaluate the most effective treatment strategy.

## Methods

**Cell culture**. Cell lines (Supplementary Table 1) were purchased from DSMZ (Germany) or ATCC (USA) and maintained in RPMI1640 + GlutaMax® (Gibco) supplemented with 10% fetal bovine serum (Gibco) and antibiotics at a density of $0.2-2 \times 10^6$ cells/ml in a humidified incubator with 5% $CO_2$ at 37 °C. Cells were periodically tested for the absence of mycoplasma contamination using the MycoAlert Mycoplasma Detection Kit (Lonza cat# LT07-118). Cell lines authentication was performed via short tandem repeat (STR) profiling.

**Western blotting**. Membranes were incubated with the following primary antibodies (1:1000 dilution, if not stated otherwise): anti-P-Tyr-1000 (Cell Signaling Technology cat# 8954), anti-Lck (Cell Signaling Technology cat# 2752), anti-Src L4A1 (Cell Signaling Technology cat# 2110), anti-phospho Lck (Tyr505) (Cell Signaling Technology cat# 2751), anti-phospho Src Family (Tyr416) (Cell Signaling Technology cat# 2101), anti-IGF1Rβ (Cell Signaling Technology cat# 3027), anti-phospho IGF1Rβ (Tyr1135) (Cell Signaling Technology cat# 3918), anti-phospho mTOR (Ser2448) (Cell Signaling Technology cat# 2971), anti-phospho p70 S6K (Thr421/Ser424) (Cell Signaling Technology cat# 9204), anti-AKT (Cell Signaling Technology cat# 9272), anti-phospho AKT (Ser473) (Cell Signaling Technology cat# 9271), anti-p44-42 MAPK (ERK1/2) (137F5) (Cell Signaling Technology cat# 4695), anti-phospho p44-42 MAPK (Thr202/Tyr204) (D13.14.4E) (Cell Signaling Technology cat# 4370), anti-cleaved caspase-3 (Asp175) (Cell Signaling Technology cat# 9661), and anti-β actin (Abcam, cat# ab6276, 1:10,000).

For protein bands staining, SDS-PAGE gels were stained using the Colloidal Blue Staining kit (Invitrogen cat# LC6025) according to the manufacturer protocol. Uncropped and unprocessed blots are provided in the Source Data file.

**Flow cytometry**. Experiments were performed using the ZE5 flow cytometer (BioRAD). For cell cycle analysis, 200,000 cells per condition were stained with Hoechst (7.5 μg/ml) for 1 h at 37 °C and then incubated for 15 min on ice before FACS analysis. For Annexin V/propidium iodide (PI) staining of apoptotic cells, 200,000 cells were stained with Annexin V-APC antibody (Biolegend cat# 640920) diluted 1:20 in Annexin V-binding buffer (Invitrogen cat# V13246) for 15 min at room temperature (RT) in the dark. PI (Miltenyi) was added at a final concentration of 0.5 μg/ml just before the FACS measurement. Data analysis was performed using FlowJo v10.7.1 (FlowJo). Examples of the sequential gatings used for the FACS data analyses are shown in Supplementary Fig. 9.

**Generation of patient-derived xenografts**. Blasts obtained from pediatric patients diagnosed with T-ALL were provided by the Dutch Childhood Oncology Group (DCOG) upon signed informed consent and in accordance with the declaration of Helsinki. Animal experiments were approved by the Animal Welfare Committee of the Princess Máxima Center for pediatric oncology (Utrecht, the Netherlands) and were carried out at the animal facility of the Hubrecht Institute (Utrecht, the Netherlands) under specific pathogen-free conditions and in accordance with animal welfare, FELASA (Federation of European Laboratory Animal Science Associations), ethical, and institutional guidelines. Mice were hosted in individually ventilated cages in groups of 2–3 mice per cage. Briefly, viably frozen human blasts were intravenously injected into immunocompromised NOD/scid/Gamma (NSG) female mice of 8–10 weeks of age (Charles River, France). Mice were constantly monitored for leukemia development and disease burden was assessed by detection of human CD45+ cells in the murine blood by tail vein cut and FACS analysis. Mice were sacrificed when presenting symptoms of leukemia (lack of grooming and activity, hunched back position, visible loss of weight) or when the circulating human CD45+ cells reached 50%. Human blasts were isolated from the murine spleen using the Lymphoprep density gradient separation (STEMCELL technologies, Canada). Purified blasts were either immediately harvested for phosphoproteomic analyses or viably frozen until further usage. The mutational status of primary cells and their related PDXs was previously investigated by whole-exome sequencing[46]. The full list of somatic mutations of the T-ALL xenografts used in this study is reported in Supplementary Data 1.

**Phosphorylated peptide enrichment and mass spectrometry analysis**. Cell lines were harvested in their exponential growth phase to preserve physiological signaling while human CD45+ blasts obtained from the murine spleen were immediately harvested after the Lymphoprep density gradient separation. Briefly, cells were spun down at $250 \times g$ for 5 min, washed in cold PBS, spun down again, and harvested in 9 M urea/20 mM HEPES (pH 8) lysis buffer containing 1 mM sodium orthovanadate, 2.5 mM sodium pyrophosphate, and 1 mM β-glycerophosphate. Cell lysates were thoroughly vortexed at maximum speed for 30 s, snap-frozen in liquid nitrogen and stored at −80 °C until further usage. Before the enrichment step, lysates were thawed, sonicated three times at 18-micron amplitude (30 s on/60 s off) using the MSE Soniprep 150 sonicator (MSE) on ice. Cleared lysates were diluted to a concentration of 2 mg/ml and 5 mg of protein input was used for each sample. Proteins were reduced with 2 mM DTT for 30 min at 55 °C, alkylated using 5 mM iodoacetamide for 15 min at RT in the dark and eventually digested overnight with Sequencing Grade Modified Trypsin (Promega cat# V5111) at RT. Digested peptides were purified using OASIS HLB Cartridges (6 cc, 500 mg Sorbent, 60 μm particle size. Waters cat# 186000115) and lyophilized. Phospho-tyrosine peptides were enriched via immunoprecipitation (IP) using the PTMScan® Phospho-Tyrosine Rabbit mAb (P-Tyr-1000) Kit (Cell Signaling Technology cat# 8803) according to the manufacturer protocol, using 4 μl of bead slurry for each mg of protein input. Phospho-tyrosine peptides were eluted in 0.15% trifluoroacetic acid (TFA) and the unbound peptide fraction was used for complementary phospho-serine and phospho-threonine peptides capturing using custom-made TiO2 C8-fitted tips. Eventually, eluted phosphorylated peptides were desalted using 20 μl SDB-XC StageTips (Prepared from Empore™ SPE Disks with SDB-XC, Sigma cat# 66884-U) prior to LC-MS analysis. For global protein expression analysis, 1 μg of total lysate was subjected to liquid chromatography-mass spectrometry (LC-MS). LC-MS analyses were performed as previously described[27]. Briefly, phosphopeptides were dried in a vacuum centrifuge and dissolved in 20 μl 0.5% trifluoroacetic acid (TFA)/4% acetonitrile (ACN) prior to injection; 18 μl was injected using partial loop injection. Peptides were separated by an Ultimate 3000 nanoLC-MS/MS system (Thermo Fisher) equipped with a 50 cm × 75 μm ID Acclaim Pepmap (C18, 1.9 μm) column. After injection, peptides were trapped at 3 μl/min on a 10 mm × 75 μm ID Acclaim Pepmap trap at 2% buffer B (buffer A: 0.1% formic acid (FA); buffer B: 80% ACN, 0.1% FA) and separated at 300 nl/min in a 10–40% buffer B gradient in 90 min (125 min inject-to-inject) at 35 °C. Eluting peptides were ionized at a potential of +2 kVa into a Q Exactive HF mass spectrometer (Thermo Fisher). Intact masses were measured from $m/z$ 350–1400 at resolution 120,000 (at $m/z$ 200) in the Orbitrap using an AGC target value of 3E6 charges and a maxIT of 100 ms. The top 15 for peptide signals (charge-states 2+ and higher) were submitted to MS/MS in the HCD (higher-energy collision) cell (1.4 amu isolation width, 26% normalized collision energy). MS/MS spectra were acquired at resolution 15,000 (at $m/z$ 200) in the Orbitrap using an AGC target value of 1E6 charges, a maxIT of 64 ms and an underfill ratio of 0.1%. This results in an intensity threshold for MS/MS of 1.3E5. Dynamic exclusion was applied with a repeat count of 1 and an exclusion time of 30 s. For peptide and protein identification, MS/MS spectra were searched against theoretical spectra from the UniProt complete human proteome FASTA file (release January 2018, 42,258 entries) using the MaxQuant 1.6.0.16 software[47] with the following settings: enzyme specificity = trypsin, missed cleavages allowed = 2, fixed modification = cysteine carboxamidomethylation; variable modification = serine, threonine and tyrosine phosphorylation, methionine oxidation, and N-terminal acetylation; MS tolerance = 4.5 ppm and MS/MS tolerance = 20 ppm. For both peptide and protein identifications, the false discovery rate was set at 1% for filtering using a decoy database strategy. The minimal peptide length was set at seven amino acids, the minimum Andromeda score for modified

peptides at 40, and the corresponding minimum delta score at 6. Moreover, the "match between runs" option was used to propagate the peptides identification across samples.

**Isolation of human thymocytes and ex vivo drug treatment**. Normal pediatric thymic tissues were obtained according to the study protocol TCbio-18-181 approved by the ethical committee and the biobank of the Utrecht university medical center (the Netherlands). Informed written consent for research purposes was provided by all the legal guardians of the participants. Briefly, after surgical removal, thymic tissue biopsies were mechanically disrupted in RPMI-1640 medium (Gibco) supplemented with fetal bovine serum (Gibco) and antibiotics to obtain a single-cell suspension. Isolated thymocytes were washed in PBS and contaminating red blood cells were removed via osmotic shock using the RBC lysis buffer (BioLegend, cat #420301) according to the manufacturer protocol. Cells were diluted to a concentration of $2.5 \times 10^6$ cells/ml and dispensed into a 384 multiwell plate (Corning) using the semi-automated Multidrop dispenser (Thermo Fisher) in duplicates. Drugs were diluted in DMSO at a concentration of 10 mM and dispensed using a TECAN D300e digital dispenser in a range of 1 nM–10 µM. Cell viability was evaluated at the time of seeding ($t = 0$) and after 72 h incubation using the CellTiter-Glo luminescence assay (Promega) according to the manufacturer protocol.

**INKA analyses**. Inference of highly active kinases from phosphoproteomic data was performed as previously described[27]. Integrative iNferred Kinase Activity (INKA) scores are calculated based on four parameters: the sum of all phosphorylated peptides belonging to a kinase; the detection of the phosphorylated kinase activation domain (kinase-centric parameters), 3) the detection of known phosphorylated substrates and the presence of predicted phosphorylated substrates (substrate-centric parameters)[22,27]. The latest version of the INKA pipeline is available online at https://inkascore.org/.

**Drug screenings**. Cytotoxicity assays were performed as previously described[48]. Alternatively, cells were seeded in triplicate in 96-multiwell plates and incubated with kinase inhibitors (Supplementary Table 2) in a concentration range from 3.2 nM to 32 µM for 72 h. For combination treatment assays in cell lines, a fixed concentration of BMS-754807, linsitinib, GSK-4529A, or ipatasertib was added to the dasatinib range. Cell viability was calculated in relation to untreated control cells using the colorimetric Thiazolyl Blue Tetrazolium Bromide (MTT) assay (Sigma-Aldrich cat# 475989). Graphs were obtained using the GraphPad Prism 9.0.1 software (GraphPad Prism, nonlinear regression, inhibitor vs. response; three parameters). Synergy upon treatment combination was calculated using the Chou–Talalay method[49], according to the following formula: combination index $(CI) = D1/Dx1 + D2/Dx2$, where Dx1 and Dx2 are the $IC_{50}$ of the single drugs while D1 and D2 are the drug concentrations achieving 50% reduction in cell viability in the combined treatment.

For synergy testing in PDXs, viably frozen T-ALL blasts purified from the murine spleen were thawed and cultured in RPMI1640 + GlutaMax® (Gibco) supplemented with 20% fetal calf serum (Gibco) and antibiotics in the absence of cytokines and feeder layer. Cells were dispensed into a 384 multiwell plate (Corning) using the semi-automated Multidrop dispenser (Thermo Fisher) in duplicates. Drugs were diluted in DMSO at a concentration of 10 mM and dispensed using a TECAN D300e digital dispenser in a range of 1 nM–10 µM. Cell viability was evaluated at the time of seeding ($t = 0$) and after 72 h incubation using the CellTiter-Glo luminescence assay (Promega) according to the manufacturer protocol. Synergy was evaluated using the SynergyFinder R package (version 2.4.16)[50] applying the Zero-Interaction Potency (ZIP) method[30]. In case of negative inhibition values, a partial correction was applied to avoid an overestimation of the synergistic effect with a combined treatment[30]. Drug combinations with a ZIP synergy score higher than 10 (corresponding to a deviation from the reference model above 10%) were considered synergistic.

**Statistical analyses**. Statistical analyses were performed via a paired, two-sided Student's $t$-test using the GraphPad Prism 9.0.1 software (GraphPad Prism). The number of biological replicates and the exact $p$-values are indicated in the figure legends.

**Reporting summary**. Further information on research design is available in the Nature Research Reporting Summary linked to this article.

## Data availability

The mass spectrometry proteomics data have been deposited to the ProteomeXchange Consortium via the PRIDE partner repository with the dataset identifier PXD024807. The human Swiss-Prot database used for raw data search was downloaded from the UniProt database (https://www.uniprot.org/). The AML phosphoproteomic data[22] used in Fig. 1e was downloaded from the ProteomeXchange Consortium using the accession code PXD007237. The targets of milciclib were identified browsing the ProteomicsDB database[28,29] (https://www.proteomicsdb.org/). Source data are provided with this paper.

## Code availability

The latest version of the INKA code used in this manuscript is available online at https://inkascore.org/.

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

## Acknowledgements

This study was supported by the Dutch Cancer Society (KWF Kankerbestrijding, grant KWF2016_10355 to V.C.) and the foundation Kinderen Kankervrij (grant Kika-295 to M.T.M. and KiKa-335 to V.M.P.). Furthermore, Cancer Center Amsterdam and the Netherlands Organization for Scientific Research (NWO Middelgroot, #91116017) are acknowledged for the support of the mass spectrometry infrastructure. The authors would like to thank dr. Bram van Wijk for providing the thymic tissue and Chris Meulenbroeks for the thymocytes isolation.

## Author contributions

V.C. designed and performed experiments, analyzed data, performed bioinformatic analyses, and wrote the manuscript. M.T.M., R.R.d.G.-d.H., and V.M.P performed experiments. R.H., A.A.H., and T.V.P. performed bioinformatic analyses. S.R.P. performed the mass spectrometry measurements and analyzed data. K.O. and A.A.F. provided samples. G.J.R.Z. provided the LCK inhibitor and expertise in drug screening. C.R.J. and J.P.P.M. designed and supervised the study and wrote the manuscript.

## Competing interests

G.J.R.Z. is founder, shareholder, and managing director of Oncolines B.V. All the other authors declare no conflict of interest.
