## [Peer Review File · Nature Communications]

Phosphoproteomic Profiling of T Cell Acute Lymphoblastic Leukemia Reveals Targetable Kinases and Combination Treatment StrategiesREVIEWER COMMENTS

Reviewer #1 (Remarks to the Author):

In this elegant study, Vantina Cordo and colleagues have performed a phosphoproteomics profiling of T-ALL to identify targetable kinases. The authors started with an exploratory unbiased profiling of a panel of 11 T-ALL cell lines. To translate their findings to clinical applications, they supplemented their profiling with drug screenings and validation in a cohort of patient-derived xenograft models.

This work is excellent, technically well performed with inclusion of necessary replicates and correct data interpretations. Their approach highlights the need for more proteomics-based approaches to identify leukemia vulnerabilities and exemplifies that it can be used to guide combination treatment strategies.

Few remarks that need to be addressed:

1) The intro/discussion should be extended with reported druggable kinases in T-ALL, like TYK2, PIM1 and Polo-like kinases (PLK) and Aurora kinases; are these also identified in the phosphoproteomics profiling? The font of the identified kinases in Figure 1B is too small.

2) The use of the 4 PDX models makes the study highly relevant. Due to the limited number of T-ALL patients samples it is logic that authors did their first exploratory phase using T-ALL cell lines. I fully agree with their discussion that in future a second more extended profiling will be highly informative by using more patients from different molecular T-ALL subtypes. At this point, I am missing details on the used PDXs in supplemental table 1, which makes it impossible to link the results with the genetic profile of these patients.

3) One of the limitations of the PDX models is that their ex vivo growth is limited, and often depend on supplemented cytokines (IL7?) or a feeder layer. If I am correct, the ex vivo culture conditions have not been included in the manuscript. The drug-response of each PDX may be highly variable because of 'sub-optimal' growth conditions and may skew the conclusion. Is this the reason why the effect of miliclib was not evaluated in the PDX models? The monotherapy effect of Dasatinib and INSR inhibitor presented in supplemental figure 8 should be included in Figure 6 as this is crucial info for the interpretation of these results. Because of the limitations (ex vivo culture conditions), further evaluation of the therapeutic effects of miliclib and dasatinib+INSR inhibition in an in vivo setting (on T-ALL PDX-02) certainly would strengthen the manuscript. Alternatively, authors should mention the limitations of their study.

4) Figure 4: the correlation between the INKA score and the response to BMS-754807 is not obvious. Not only two cell lines with high INKA score, SUP-T1 and Jurkat, are not responsive; also two other cell lines that have low INKA score, PEER and CCRF-CEM, are responsive. How specific is this inhibitor? Also the synergistic effect with the dasatinib+IGF1R in supplemental Figure 7 is not convincing. These data should be confirmed with a second IGF1R/INSR inhibitor or via (drug-inducible) genetic depletion.

Reviewer #2 (Remarks to the Author):

The authors of "Phosphoproteomic profiling of T Cell acute lymphoblastic leukemia reveals targetable kinases and combination treatment strategies" have employed mass spectrometry-based global phosphoproteomic profiling using a panel T-ALL cell lines to identify targetable kinases. Herein they identify an impressive 21,000 phosphosites on 4,896 phosphoproteins, including 217 kinases. The application of phosphoproteomics using clinically relevant patient samples to aid in clinical decision making, is the holy grail of clinical proteomics for cancer patients. The authors have established bioinformatic pipelines to help deconvolute these complex phosphoproteomes and employ them to reveal SRC family kinases and cyclin-dependent kinase show ubiquitous activity in T-ALL cell lines.

The article is well written however, I have several key issues with the manuscript.

The authors omit key T-ALL references such as references to mutations in the tyrosine-protein kinase JAK3 seen 16% of T-ALL cases PMID: 26206799 and conclude that no other global, MS-based phosphoproteomic study have been published to predict drug sensitivity, however, miss this study PMID: 28852199.

The authors should list known genomic drivers of these cell lines in Supplementary Table S1, so correlation between selection of effective therapies based on phosphoproteomic signatures differ from those that would be encouraged based on genomics.

There are no statistics in Supplementary Figure 1?

It is not overly clear to me how the enriched phosphopeptides were quantitated across samples? It is also unclear whether biological or technical replicates were performed except in LOUCY and CCRF-CEM performed in duplicate, hence statistical analysis comparing these cell lines is impossible. I also find it troubling that there are no controls cell lines or tissue. This is highlighted by the rapamycin results which are highly effective across cell lines but known to be highly immunosuppressive in any case.

I find it challenging to use the data in Fig 1 to come to my own conclusions.

Given the priorities of assessment of the tyrosine phosphoproteome, I am unclear as to why the authors start their results by dissection of the serine/threonine phosphoproteome, which yields little in the way of a therapeutic vulnerabilities.

All of Figure 2 requires corroborative statistical analysis. Is the difference (statistical) in viability (A/V+) HSB-2 cells treated with 100 nM Milciclib?

It is an interesting choice to assess substrate-level phosphorylation after 3-day treatment with Dasatinib in Figure 3? Authors need to include an assessment of cell viability markers to show on-target effects or whether this is general cell death.

Previous studies also report the benefit of combined SFK and IGF-1R inhibition in vitro and in vivo. Indeed, these studies also show upon IGF-1R inhibition, increased activity of SFK, highlighting the potential of the combination.

Although the manuscript tests the efficacy of therapies following a repeat of INKA analysis using isolated blasts ex vivo, what is missing are studies to assess the preclinical utility of the pipeline/results.

We thank the reviewers for the careful and detailed evaluation of our manuscript and the appreciation of our work. We have addressed the points raised by the reviewers as outlined in the *point-by-point* response below. Additionally, we have highlighted in yellow the alterations in the text, figures, and legends of the original manuscript. Moreover, we provide here additional data to answer some of the points raised by the reviewers (“*Figures for reviewers*”).

REVIEWER COMMENTS

Reviewer #1 (Remarks to the Author):

In this elegant study, Valentina Cordo and colleagues have performed a phosphoproteomics profiling of T-ALL to identify targetable kinases. The authors started with an exploratory unbiased profiling of a panel of 11 T-ALL cell lines. To translate their findings to clinical applications, they supplemented their profiling with drug screenings and validation in a cohort of patient-derived xenograft models.

This work is excellent, technically well performed with inclusion of necessary replicates and correct data interpretations. Their approach highlights the need for more proteomics-based approaches to identify leukemia vulnerabilities and exemplifies that it can be used to guide combination treatment strategies.

Few remarks that need to be addressed:

1) The intro/discussion should be extended with reported druggable kinases in T-ALL, like TYK2, PIM1 and Polo-like kinases (PLK) and Aurora kinases; are these also identified in the phosphoproteomics profiling? The font of the identified kinases in Figure 1B is too small.

We thank the reviewer for the appreciation of our elaborate work and for pointing out some interesting kinases that were missing in our introduction/discussion. Therefore, we added text and references indicating the role of TYK2, PIM1, PLKs and AURK in T-ALL and their possible targeting in the introduction section (lines 62-66 and ref #8-13).

In our phosphoproteome profiling, we identify a low TYK2 kinase activity in the cell lines (HSB-2, PEER, ALL-SIL, HPB-ALL, and CCRF-CEM) and in the four PDXs as reported in Figure 1B (ranking 14th-20th in the Top20-tyrosine INKA plots) and Figure 6B (ranking 12th-20th in the Top20-tyrosine INKA plots), respectively. Very low PIM1, PLK1 and AURKA activity was inferred (ranking beyond the Top20 active kinases in each sample and were therefore not depicted in the plots).

To improve the readability of Figures 1B and 1C, we increased the font of the text in the INKA plots.

2) The use of the 4 PDX models makes the study highly relevant. Due to the limited number of T-ALL patients samples it is logic that authors did their first exploratory phase using T-ALL cell lines. I fully agree with their discussion that in future a second more extended profiling will be highly informative by using more patients from different molecular T-ALL subtypes. At this point, I am missing details on the used PDXs in supplemental table 1, which makes it impossible to link the results with the genetic profile of these patients.

We agree with the reviewer that the missing genomic information is important for the correct interpretation and analysis of our results. Therefore, we added an additional supplementary table (Supplementary Table 3) listing all the somatic mutations identified in the 4 PDXs by whole-exome sequencing. Moreover, we highlight in the results and discussion sections the relation between the lack of somatic mutations in *INSR* and *IGF1R* and the detection of active INSR/IGF-1R in lines 252-254 and lines 308-311. In addition, the presence of JAK mutations in PDX-04 and the high JAK activity inferred, the sensitivity to ruxolitinib, and the synergy between ruxolitinib and dasatinib are

discussed in lines 259-260, lines 291-293 and lines 324-325.

3) One of the limitations of the PDX models is that their ex vivo growth is limited, and often depend on supplemented cytokines (IL7?) or a feeder layer. If I am correct, the ex vivo culture conditions have not been included in the manuscript. The drug-response of each PDX may be highly variable because of 'sub-optimal' growth conditions and may skew the conclusion. Is this the reason why the effect of milciclib was not evaluated in the PDX models?

We thank the reviewer for the detailed analysis of our PDX-based drug screening results.

Viably frozen purified blasts from T-ALL PDXs were thawed and cultured in RPMI1640+Glutamax supplemented with 20% fetal calf serum and antibiotics for 3 days, in the absence of cytokines and feeder layer to reduce the complexity of our experimental settings. In fact, the addition of cytokines such as IL7 could alter the intracellular signaling and promote therapy resistance as previously described (Delgado-Martin *et al.*, Leukemia 2017; van der Zwet *et al.*, Leukemia 2021). We now added a more detailed description of our experimental settings in the Material and Methods section at lines 437-439.

We initially did not include the evaluation of milciclib sensitivity in PDXs since we wanted to focus on the validation of our strategy for the identification of INKA-guided effective treatment combinations from phospho-tyrosine data (the dataset that provides the LCK, SRC, INSR/IGF-1R activities already investigated in the cell lines panel). Nevertheless, we tested the sensitivity of the 4 PDXs to milciclib treatment *ex vivo* and we also evaluated the cell survival/proliferation, as illustrated below and in Supplementary Figures 5G-H. The 4 PDXs had a different survival and proliferation rate during 72 hours of culture *in vitro* in the absence of cytokines and feeder layer. While untreated PDX-01 and PDX-04 cells had a decrease in survival (decrease in mean luminescence signal at day 3 compared to day 0), PDX-02 cells survived well *in vitro* and PDX-03 showed even a clear proliferation during 72 hours of culture *in vitro* (panel G). Panel H illustrates the dose-response curves for milciclib treatment. All the PDXs showed sensitivity to milciclib with IC50 values lower than 200nM and included in the clinical range (lower than the maximum plasma concentration achieved in patients, green box). It is important to notice that PDX-03, which showed active proliferation *in vitro* seems to be the most sensitive to milciclib treatment. We agree with the reviewer that sub-optimal culturing condition could affect the results of the drug screening (in particular when testing chemotherapeutics or drugs that interfere with the cell cycle). Nevertheless, this seems not to be the case for milciclib in our limited PDX cohort since we did not see resistance to treatment in the absence of cell proliferation.

Supplementary Figure 5

G

H

Supplementary Figure 5G-H: T-ALL PDXs are sensitive to milciclib treatment *ex vivo*. G: Luminescence signal recorded at day 0 (time of seeding) and day 3 (72 h culture). Symbols indicate replicates per conditions (N = 4), the horizontal line the mean, and the error bars indicate the average of the mean. H: dose-response curve of milciclib treatment *ex vivo* in 4 PDXs. Cells were treated with increasing concentrations of milciclib (1nM-10 μ M) for 72 hours and the survival was calculated in comparison to the untreated control (DMSO only). The green box indicates the clinical concentration range of milciclib.

The monotherapy effect of Dasatinib and INSR inhibitor presented in supplemental figure 8 should be included in Figure 6 as this is crucial info for the interpretation of these results.

In agreement with the reviewer, we now moved the monotherapy dose-response curves in panels C and E of Figure 6, respectively.

Because of the limitations (ex vivo culture conditions), further evaluation of the therapeutic effects of milciclib and dasatinib+INSR inhibition in an in vivo setting (on T-ALL PDX-02) certainly would strengthen the manuscript. Alternatively, authors should mention the limitations of their study.

We fully agree with the reviewer on the limitations of our drug screening approach. We also agree that *in vivo* data would certainly strengthen the validity of our findings. However, the main aim of this study was to propose phosphoproteomics as useful tool to identify non-genomic targets and to guide the testing of novel treatment strategies, rather than validating a single treatment option. Therefore, now we list and discuss the limitations of our study (such as the *in vitro* drug screening in the absence of active cell proliferation that might affect sensitivity to some drugs, the need to expand the limited PDXs cohort, the need for *in vivo* validations of effective treatment strategies) in the discussion (lines 332-338).

4) Figure 4: the correlation between the INKA score and the response to BMS-754807 is not obvious. Not only two cell lines with high INKA score, SUP-T1 and Jurkat, are not responsive; also two other cell lines that have low INKA score, PEER and CCRF-CEM, are responsive. How specific is this inhibitor?

We agree with the reviewer that the correlation between the INKA score and the BMS-754807 response might not be immediate. First, the absolute INKA score for INSR/IGF-1R should be considered within a single sample (therefore in the context of the Top20 active kinases for each cell line). Figure 4A is used to show that most of the T-ALL cell lines have active INSR/IGF-1R. As illustrated in the *Figure 1 for Reviewers* below, we took Figure 4B from the manuscript and highlighted in orange the lines with low/no INKA score for INSR/IGF-1R such as PEER, CCRF-CEM and HSB-2 (panel A). Differently from the most sensitive lines highlighted in blue (ALL-SIL, MOLT-16 and HPB-ALL, IC50 values < 250nM), PEER, CCRF-CEM and HSB-2 have IC50 values around 1 μ M. We therefore checked the selectivity of BMS-754807 and all the possible targets inhibited by the drug at different concentrations (dataset from Klaeger *et al.*, The target landscape of clinical kinase drugs, Science 2017) and we report in the figure below (panel B) the list of BMS-754807 targets and their activity inhibition at 100nM, 1 μ M, and 2 μ M. As reported by Klaeger and colleagues, kinase inhibitors lose their specificity when increasing the drug concentration. In fact, BMS-754807 has only 3 predicted targets (inhibition > 50%) at 100nM while the number of kinases inhibited (inhibition > 50%) goes up to 11 (1 μ M) and 15 (2 μ M) at increasing concentrations. Therefore, we cannot exclude

that the response obtained with BMS-754807 concentrations above 1 μM are caused by off-target inhibition of additional kinases rather than INSR and IGF-1R only. In our study, we generally consider cell lines sensitive to a kinase inhibitor when the IC50 values are in the nanomolar range ($< 1\mu\text{M}$) to avoid misinterpreting an off-target kinase inhibition/general cytotoxicity with an actual on-target effect. Indeed, we consider only ALL-SIL, HPB-ALL and MOLT-16 as sensitive cell lines as indicated in the text at lines 203-205. For the same reason we decided to use only the IC20 values of BMS-754807 (30-300nM) for the drug combination screenings in the cell lines.

Figure 1 for reviewers

Figure 1 for reviewers: T-ALL cell lines show different sensitivity to the INSR/IGF-1R inhibitor BMS-754807. A) Dose-response curves of BMS-754807 treatment in 11 T-ALL cell lines. Cells were treated with increasing concentrations of the INSR/IGF-1R inhibitor BMS-754807 (3.2nM–32 μM range) and cell viability was assessed after 72 hours in comparison to the untreated control (DMSO only). Blue: very sensitive lines. Red: resistant lines. Orange: lines with low or no INKA score for INSR/IGF-1R. B) Predicted targets of BMS-75487 and relative kinase inhibition values at 100nM, 1 μM , and 2 μM . Red: main targets (INSR and IGF-1R). Yellow: targets with kinase activity inhibition higher than 50% (source: Klaeger *et al.*, 2017; Samaras *et al.*, 2019; <https://www.proteomicsdb.org/>).

Also the synergistic effect with the dasatinib+IGF1R in supplemental Figure 7 is not convincing. These data should be confirmed with a second IGF1R/INSR inhibitor or via (drug-inducible) genetic depletion.

We agree with the reviewer regarding the need of further validation of the synergistic effect of dasatinib and an INSR/IGF-1R inhibitor. Therefore, we tested two other INSR/IGF-1R inhibitors, linsitinib (OSI-906) and GSK1904529A (GSK-4529A), respectively. We confirmed the synergistic effect of combined SRC/LCK and INSR/IGF-1R inhibition when treating cells with dasatinib and a fixed concentration (IC20) of linsitinib (500nM) and GSK-4529A (60nM). We therefore removed the graph in Supplementary Figure 7 (which now illustrates the dose-response curves to the single INSR/IGF-1R inhibitors and their corresponding IC20 values) and added the two additional dasatinib + INSR/IGF-1Ri combinations in Figure 4E of the manuscript, demonstrating synergistic effects of all 3 INSR/IGF-1R inhibitors when combined with dasatinib. The text at lines 213-218 in the results section was accordingly changed.

Reviewer #2 (Remarks to the Author):

The authors of “Phosphoproteomic profiling of T Cell acute lymphoblastic leukemia reveals targetable kinases and combination treatment strategies” have employed mass spectrometry-based global phosphoproteomic profiling using a panel T-ALL cell lines to identify targetable kinases. Herein they identify an impressive 21,000 phosphosites on 4,896 phosphoproteins, including 217 kinases.

The application of phosphoproteomics using clinically relevant patient samples to aid in clinical decision making, is the holy grail of clinical proteomics for cancer patients. The authors have established bioinformatic pipelines to help deconvolute these complex phosphoproteomes and employ them to reveal SRC family kinases and cyclin-dependent kinase show ubiquitous activity in T-ALL cell lines.

The article is well written however, I have several key issues with the manuscript.

The authors omit key T-ALL references such as references to mutations in the tyrosine-protein kinase JAK3 seen 16% of T-ALL cases PMID: 26206799 and conclude that no other global, MS-based phosphoproteomic study have been published to predict drug sensitivity, however, miss this study PMID: 28852199.

We thank the reviewer for the insightful suggestions. We added the recommended references in the introduction section of the manuscript at lines 60 (ref #6) and lines 83-84 (ref #25).

The authors should list known genomic drivers of these cell lines in Supplementary Table S1, so correlation between selection of effective therapies based on phosphoproteomic signatures differ from those that would be encouraged based on genomics.

We agree with the reviewer that genomic information of the cell lines used is necessary to highlight the differences of putative targets suggested by the phosphoproteomic analyses. Therefore, we now added the subtype, the genomic drivers and known mutated oncogenes of the 11 cell lines in Supplementary Table 1.

There are no statistics in Supplementary Figure 1?

Regarding to the lack of statistics in Supplementary Figure 1A, we would like to clarify that the graph only shows the number of phosphorylated peptides recovered and identified for each cell line in a single mass spectrometry measurement. To exclude that the visible higher recovery of phosphorylated peptides in HSB-2 could be due to a possible error, a higher total lysate input or a different efficiency of purification during the phospho-tyrosine immunoprecipitation, we checked the total phosphorylation level prior to any phospho-enrichment via western blotting (anti-phosphotyrosine antibody) as shown in Supplementary Figure 1C. Despite an equal protein loading (blue Coomassie staining in Supplementary Figure 1C-right), HSB-2 cells show a higher phospho-tyrosine signal compared to any other cell line already before any phospho-enrichment. Therefore, we concluded that HSB-2 cells present an enhanced phospho-tyrosine signaling (possibly due to the *TCR β -LCK* translocation that results in higher LCK expression) as intrinsic characteristic. We did not perform any statistical analysis to check whether the higher recovery of phosphorylated peptides was significantly different in comparison to the other lines.

It is not overly clear to me how the enriched phosphopeptides were quantitated across samples?

Label-free mass spectrometry (MS) measurements with ion intensity-based relative quantification were performed as previously described (Piersma SR *et al.*, Journal of Proteomics, 2015; Beekhof R. *et al.*, Molecular Systems Biology, 2019).

We added the following details to the supplementary methods section (page 1-2, line 15-31): Mass-spectrometry measurements were performed using data-dependent acquisition MS. Every scan cycle of ~1 sec starts with measurement of intact peptide masses (MS1) and subsequently the top15 highest peptide signals are sequentially isolated in the quadrupole and fragmented in the HCD collision cell to acquire MS/MS spectra. For peptide and protein identification, MS1 and MS/MS spectra were searched against the Swissprot complete human proteome FASTA file (release January 2018, 42,258 entries) using the MaxQuant 1.6.0.16 software and applying a false discovery rate (FDR) cut-off of 1% at the peptide, phosphosite and protein level.

Phosphopeptides and phosphosites were quantified from the MS1 Intensities of the eluting phosphopeptides. Phosphopeptide intensities were used by MaxQuant to calculate phosphosite intensities. The relevant information can be extracted from the phospho(STY)sites.txt and modificationSpecificPeptides.txt files produced by MaxQuant. The quantification of proteins was performed using the MaxLFQ algorithm (Cox J. *et al.*, Mol Cell Proteomics 2014).

It is also unclear whether biological or technical replicates were performed except in LOUCY and CCRF-CEM performed in duplicate, hence statistical analysis comparing these cell lines is impossible. I also find it troubling that there are no controls cell lines or tissue. This is highlighted by the rapamycin results which are highly effective across cell lines but known to be highly immunosuppressive in any case.

We apologize if our experimental setup was not clearly explained. For the serine/threonine phosphoproteome (TiO2 dataset), workflow duplicates (technical replicates) were used while for the phospho-tyrosine immunoprecipitation that requires a relatively high protein input (pTyr dataset), the analysis was performed on a single sample per cell line, except for CCRF-CEM and LOUCY for which two biological replicates were used as reproducibility controls to show the consistency of the dataset. We show high correlation between the biological (Figure 1D) and workflow replicates (Supplementary Figure 4) indicating robustness of the data. Additionally, in the unsupervised clustering of the pTyr-peptides illustrated in Supplementary Figure 1B, the CCRF-CEM and LOUCY biological replicates cluster together. To assure that the phospho-enrichment and the mass spectrometry measurements are correctly performed, we always include an additional reference sample as internal control for each experiment (protein lysate and phospho-peptides enriched fractions from the colorectal cancer cell line HCT-116, previously characterized by Piersma SR *et al.*, Journal of Proteomics, 2015 and Labots M *et al.*, Journal of Proteomics, 2017; data not shown). Our workflow reproducibility is high (Piersma SR *et al.*, Journal of Proteomics, 2015, Van der Mijn JC *et al.*, International Journal of Cancer 2016; Rolfs F *et al.*, Molecular & Cellular Proteomics, 2021) therefore the inclusion of biological replicates of cell lines samples does not add significant information. Moreover, our aim was to show how phosphoproteomics can be applied to a single sample to investigate putative targets in an experimental pipeline that can be easily translated into the clinical setting (patient-tailored treatment). In fact, given the limited availability of patient blasts at diagnosis (or during treatment monitoring/relapse) and given the need for a large amount of protein input for an in-depth phospho-tyrosine immunoprecipitation (4-5mg protein input,

corresponding to at least 250 million cells per sample), it would be very difficult to perform such type of analyses in triplicate when using primary cells or patient-derived xenografts.

The results of the drug screening performed in this study are obtained in relation to untreated cells (DMSO only-treated cells) after 3 days of treatment. We did not include any additional cell line or tissue since we did not investigate the phosphoproteome of healthy cells/tissue (*e.g.*, T-cells from healthy donors) in comparison with T-ALL since this goes beyond the scope of this study. Moreover, most of the inhibitors used in this study (*e.g.*, imatinib, dasatinib, milciclib, everolimus, and ruxolitinib) are already used in the clinical setting (either as approved therapy for other cancer types or currently investigated in clinical trials, also in the context of relapsed T-ALL) therefore the off-target effects and the related possible toxicities of these drugs are known. Nevertheless, we tested the sensitivity of human thymocytes freshly isolated from a healthy pediatric thymus to the drugs used in this manuscript such as dasatinib, milciclib, BMS-754807, ruxolitinib and rapamycin (everolimus). We did not detect any effect during a 3-day *ex vivo* treatment with rapamycin, as illustrated in Supplementary Figure 5E and the *Figure 2 for reviewers* below. Only for milciclib we saw a decrease in cell survival upon treatment but only at concentrations above 1 μ M while T-ALL cells lines (and PDXs, see Supplementary Figure 5H, in relation to another comment on the drug screenings from Reviewer #1) showed sensitivity at lower concentrations.

Figure 2 for reviewers

I find it challenging to use the data in Fig 1 to come to my own conclusions.

We would like to clarify that the data presented in Figure 1B and 1C provide an overview of the kinase activities inferred from phosphoproteomic data (pTyr and TiO2 for phospho-tyrosine and phospho-serine/threonine kinases, respectively) and illustrates the single top20-kinase profile for each cell line separately. The aim of these two figure panels is to illustrate the active signaling nodes in the T-ALL cell lines, also highlighting the role of genomic-driven kinase activities (*e.g.*, ABL1 in *NUP214-ABL1* positive cells), when present. We agree that it can be difficult to draw immediate conclusions from the data presented in Figure 1, thus later in the manuscript we focused on selected relevant kinase activities detected in multiple cell lines and the possibility of targeting those kinases separately.

Given the priorities of assessment of the tyrosine phosphoproteome, I am unclear as to why the authors start their results by dissection of the serine/threonine phosphoproteome, which yields little in the way of a therapeutic vulnerabilities.

We agree with the reviewer on the limited therapeutic vulnerabilities highlighted by the serine/threonine phosphoproteome in our study. However, our aim was to give first a complete overview of active kinases and signaling pathways in T-ALL in general, and then dissecting the relevant signaling nodes to select the most interesting therapeutical targets. Therefore, given the lack of such a systematic, unbiased analysis of active kinase signaling in T-ALL, we think that the serine/threonine dataset is also a valuable (introductory) part of our study and a resource for the field. Therefore, we start the results with an overall description of our global strategy, highlighting the relevant kinases found in both datasets (lines 96-124). Subsequently, we briefly describe the limited therapeutical vulnerabilities highlighted by the TiOx dataset (lines 146-156) and finally analyze the targeting of CDK1 and CDK2.

All of Figure 2 requires corroborative statistical analysis. Is the difference (statistical) in viability (A/V+) HSB-2 cells treated with 100 nM Milciclib?

We agree with the reviewer that statistical analyses were missing in Figure 2. Since all the experiment were performed using biological replicates, we now added statistical significance (p values from two-tailed paired t-test for treated versus untreated cells) in Figure 2B and 2C.

The difference in the percentage of apoptotic cells between untreated HSB-2 cells and cells treated with 100nM milciclib (Figure 2C) is significantly different ($p < 0.05$ with two-tailed paired Student's t-test).

It is an interesting choice to assess substrate-level phosphorylation after 3-day treatment with Dasatinib in Figure 3? Authors need to include an assessment of cell viability markers to show on-target effects or whether this is general cell death.

We apologize if the rationale for investigating the substrate-level phosphorylation after 3-day treatment with dasatinib in Figure 3C was not clear. Since the drug screening with different ABL/SRC family kinase inhibitors was performed during 72h (Figure 3A) and given the low sensitivity for 9 out of 11 lines to dasatinib treatment, we sought to investigate the cellular mechanism of dasatinib resistance (*e.g.*, target downregulation or insufficient on-target effect). We chose the 100nM concentration for western blotting analysis since it is a clinically relevant concentration, it is enough to induce cell death in the sensitive lines HSB-2 and ALL-SIL but does not affect the viability of the other cell lines. As shown in Figure 3A, viability of 100nM dasatinib-treated cells after 3 days compared to the untreated control at day 3 (DMSO only-treated cells) is above 80% for all cell lines except HSB-2 and ALL-SIL. Thus, the results shown in Figure 3C should be correlated to the survival (% cell survival upon treatment compared to DMSO only-treated cells after 72h incubation) shown in the dose-response curves in Figure 3A. We adapted the text at lines 187-190 to better highlight the connection between panel A and C.

To confirm that despite target inhibition (decrease in pLCK-Y505 and pSRC family-Y416), there is no induction of cell death, we added cleaved caspase-3 detection to the western blot shown in Figure 3C of the manuscript. As shown here below and in Supplementary Figure 6F of the manuscript, despite a marked decrease in pLCK-Y505 and pSRC family-Y416, no cleaved caspase-3 was detected in dasatinib-resistant lines. On the contrary, when comparing a dasatinib-sensitive cell line (HSB-2) and a dasatinib-resistant line (P12-ICHIKAWA) after only 16h of 100nM dasatinib-treatment, a visible caspase-3 cleavage was detected in HSB-2 cells (Supplementary Figure 6G). We chose the 16h treatment for HSB-2 cells because is the longest that can survive *in vitro* upon 100nM dasatinib

treatment. On-target effect of the different ABL/SRC family inhibitors can also be appreciated when comparing the response of HSB-2 cells to ABL-only inhibitors (imatinib, nilotinib) to the response to broader SRC/LCK inhibitors. In fact, HSB-2 cells are resistant to the ABL-only inhibitors imatinib and nilotinib ($IC_{50} > 1\mu M$) while the IC_{50} values for the SRC/LCK inhibitors dasatinib, bosutinib, ponatinib, and the LCK inhibitor A-420983 are lower than 100nM (Figure 3A).

Supplementary Figure 6

Supplementary Figure 6: T-ALL cell lines show limited sensitivity to SFKs inhibition *in vitro*. F) Western blot analysis upon dasatinib treatment. Cell lines expressing high levels of LCK and/or SRC were treated with 100nM dasatinib for 72 hours. G) Western blot analysis upon dasatinib treatment. One dasatinib-sensitive cell line (HSB-2) and one dasatinib-resistant cell line (P12-ICHIKAWA) were treated with 100nM dasatinib for 16 hours.

Previous studies also report the benefit of combined SFK and IGF-1R inhibition *in vitro* and *in vivo*. Indeed, these studies also show upon IGF-1R inhibition, increased activity of SFK, highlighting the potential of the combination.

Although the manuscript tests the efficacy of therapies following a repeat of INKA analysis using isolated blasts *ex vivo*, what is missing are studies to assess the preclinical utility of the pipeline/results.

We thank the reviewer for the critical evaluation of our results. In relation to a similar comment from reviewer #1, we fully agree with the reviewers on the limitations of our phosphoproteomic screening approach. We also agree that *in vivo* validation would certainly strengthen the validity of relevant clinical findings. However, the main aim of this study was to propose phosphoproteomics as useful tool to identify non-genomic targets and to guide the testing of novel treatment strategies, rather than validating a single treatment option. Therefore, now we list and discuss the limitations of our study (such as the *in vitro* drug screening in the absence of active cell proliferation that might affect sensitivity to some drugs, the need to expand the limited PDXs cohort to include different disease subtypes, the need for *in vivo* validations of effective treatment strategies) in the discussion of our manuscript (lines 332-338).

The preclinical utility of the INKA-based treatments selection has been thoroughly assessed in the original methodological paper describing the pipeline (Beekhof, R. *et al.* INKA, an integrative data analysis pipeline for phosphoproteomic inference of active kinases. *Mol Syst Biol*, 2019). In this study, INKA prediction of active kinases was validated in cell lines with driving kinase oncogenes, cell lines with differential settings (*e.g.*, wild-type versus mutant lines; untreated versus treated samples), pre-treatment and on-treatment tumor needle biopsies, and patient-derived tumor xenografts. In these different analyses, INKA outperforms not only its single components but also other published kinase-activity inferring (*e.g.*, KARP method based on the detection of phosphorylated substrates, Wilkes EH *et al.*, Empirical inference of circuitry and plasticity in a kinase signaling network. *Proc Natl Acad Sci USA*, 2015). Moreover, the application of phosphoproteomic profiling and INKA-based kinase ranking for the selection of relevant treatment strategies have been already applied and in the context of solid tumors (Le Large TYS *et al.*, *J Exp Clin Cancer Res* 2021) and in acute myeloid leukemia (van Alphen CR *et al.*, *Mol Cell Proteomics* 2020; Cucchi DGJ *et al.*, *Hemasphere* 2021) indicating the value of the analysis pipeline and the relevance of the results. We now refer to the preclinical validation of the INKA pipeline also in the discussion in lines 331-332.

REVIEWER COMMENTS

Reviewer #1 (Remarks to the Author):

The authors properly addressed all of my comments regarding the previous version of their manuscript. The authors also now mention the limitations of their study in the discussion section. Accordingly, their manuscript significantly improved and I have no further concerns.

Reviewer #2 (Remarks to the Author):

I thank the authors for addressing each of my comments. The additional stringency of the new analysis improves the manuscript.

The combined effects dasatinib and BMS-754807 and dasatinib and ruxolitinib in PDX blasts ex vivo (Fig. 6C and E, respectively) need to be included on the sensitivity pots.

The authors nicely highlight that phosphoproteomics is useful for the identification of non-genomic targets to guide the testing of novel treatment strategies; however, preclinical validation including patient-derived animal models is required. This is not only to assess the efficacy, but the contribution of the bone marrow and other supporting organ tissues surrounding blasts, but also to assess toxicities, toxicities that might include hyperglycemia, skin irritation and mucositis limiting the clinical utility.

REVIEWERS' COMMENTS

Reviewer #1 (Remarks to the Author):

The authors properly addressed all of my comments regarding the previous version of their manuscript. The authors also now mention the limitations of their study in the discussion section. Accordingly, their manuscript significantly improved and I have no further concerns.

We thank the reviewer for the appreciation of our revised work, and we are delighted to see that there are no further concerns.

Reviewer #2 (Remarks to the Author):

I thank the authors for addressing each of my comments. The additional stringency of the new analysis improves the manuscript.

We thank the reviewer for acknowledging the improvements of our revised manuscript.

The combined effects dasatinib and BMS-754807 and dastinib and ruxolitinib in PDX blasts ex vivo (Fig. 6C and E, respectively) need to be included on the sensitivity pots.

We now added additional supplementary figures (Supplementary Figures 7c and 7d) showing the combined synergistic effect of the drug combinations *ex vivo*.

The authors nicely highlight that phosphoproteomics is useful for the identification of non-genomic targets to guide the testing of novel treatment strategies; however, preclinical validation including patient-derived animal models is required. This is not only to assess the efficacy, but the contribution of the bone marrow and other supporting organ tissues surrounding blasts, but also to assess toxicities, toxicities that might include hyperglycemia, skin irritation and mucositis limiting the clinical utility.

We agree with the reviewer regarding the need of *in vivo* validation of the drug combinations to investigate not only their efficacy, but also possible toxicities and possible insurgence of therapy resistance promoted by the microenvironment. Therefore, we now address these arguments in the discussion section at lines 352-355: " Further *in vivo* investigations of selected drug combinations should address not only the efficacy and the tolerability of these treatments (*i.e.*, toxicities) but also the role of the microenvironmental niches that can support blasts growth and survival. Such investigations could allow the direct translation of the preclinical findings to the clinical settings."